# An acquired scaffolding function of the DNAJ-PKAc fusion contributes to oncogenic signaling in fibrolamellar carcinoma

Rigney E Turnham[1†], F Donelson Smith[1†], Heidi L Kenerson[2‡], Mitchell H Omar[1‡], Martin Golkowski[1], Irvin Garcia[1], Renay Bauer[2], Ho-Tak Lau[1], Kevin M Sullivan[2], Lorene K Langeberg[1], Shao-En Ong[1], Kimberly J Riehle[2], Raymond S Yeung[2], John D Scott[1]*

[1]Department of Pharmacology, University of Washington Medical Center, Seattle, United States; [2]Department of Surgery, University of Washington Medical Center, Seattle, United States

**\*For correspondence:**
scottjdw@uw.edu

[†]These authors contributed equally to this work
[‡]These authors also contributed equally to this work

**Competing interests:** The authors declare that no competing interests exist.

**Abstract** Fibrolamellar carcinoma (FLC) is a rare liver cancer. FLCs uniquely produce DNAJ-PKAc, a chimeric enzyme consisting of a chaperonin-binding domain fused to the Cα subunit of protein kinase A. Biochemical analyses of clinical samples reveal that a unique property of this fusion enzyme is the ability to recruit heat shock protein 70 (Hsp70). This cellular chaperonin is frequently up-regulated in cancers. Gene-editing of mouse hepatocytes generated disease-relevant AML12[DNAJ-PKAc] cell lines. Further analyses indicate that the proto-oncogene A-kinase anchoring protein-Lbc is up-regulated in FLC and functions to cluster DNAJ-PKAc/Hsp70 sub-complexes with a RAF-MEK-ERK kinase module. Drug screening reveals Hsp70 and MEK inhibitor combinations that selectively block proliferation of AML12[DNAJ-PKAc] cells. Phosphoproteomic profiling demonstrates that DNAJ-PKAc biases the signaling landscape toward ERK activation and engages downstream kinase cascades. Thus, the oncogenic action of DNAJ-PKAc involves an acquired scaffolding function that permits recruitment of Hsp70 and mobilization of local ERK signaling.
DOI: https://doi.org/10.7554/eLife.44187.001

## Introduction

Fibrolamellar carcinoma (FLC) is a variant of liver cancer that has distinctive histologic features (*Craig et al., 1980*). This rare cancer afflicts healthy adolescents and young adults between the ages of 15–25 with no history of liver disease. This latter feature can compromise early diagnosis of FLC as patients frequently present with vague symptoms that include abdominal pain, loss of appetite, or a palpable mass. The diagnosis is often made after disease has spread outside the liver, leading to an overall survival of 35% (*Ang et al., 2013*). Unfortunately, FLC frequently recurs, as it is intractable to standard chemotherapies and radiation. Surgical resection is currently the only opportunity for a cure. The search for new therapies for these patients is hindered by the limited availability of clinical samples and a lack of disease relevant cell lines or animal models that faithfully recapitulate the pathogenesis of FLC (*Dinh et al., 2017*; *Engelholm et al., 2017*; *Kastenhuber et al., 2017*; *Oikawa et al., 2015*).

Recent transformative advances in our understanding of the molecular basis of FLC offer renewed hope for the development of drug therapies to treat this disease (*Honeyman et al., 2014*). Sequencing tumor genomes of FLCs identified the underlying genetic defect as a heterozygous in-frame deletion of ~400 kb in chromosome 19 (*Honeyman et al., 2014*; *Xu et al., 2015*). This genetic lesion

**eLife digest** Fibrolamellar carcinoma (or FLC for short) is a rare type of liver cancer that affects teenagers and young adults. FLC tumors are often resistant to standard radiotherapy or chemotherapy treatments. The only way to treat FLC is to remove tumors by surgery. However, often the tumors come back after initial treatment and spread to other locations. Therefore, there is a genuine need to improve the treatment options available to FLC patients.

The tumor cells of FLC patients contain a genetic defect that fuses together two genes, which produce proteins called DNAJ and PKAc. Normally, DNAJ helps other proteins in the cell to fold into their correct shapes, while PKAc is an enzyme that can control how cells communicate. However, it is not clear what the abnormal DNAJ-PKAc fusion protein does, or how it causes FLC.

Turnham, Smith et al. have now used gene editing to make mouse liver cells that mimic the human FLC mutation. Biochemical experiments on these cells showed that the DNAJ-PKAc protein brings together unique combinations of enzymes that drive uncontrolled cell growth. Analyzing cells taken from tumors in FLC patients confirmed that these enzymes are also activated in the human disease. Turnham, Smith et al. also found that combinations of drugs that simultaneously target the DNAJ-PKAc protein and the recruited enzymes slowed down the growth of FLC cells. More experiments are now needed to test these drug combinations on human FLC cells or in mice.
DOI: https://doi.org/10.7554/eLife.44187.002

leads to translation of a de novo chimeric gene product where the chaperonin-binding domain of heat shock protein 40 (DNAJ) is fused to the Cα subunit of PKA (*Cheung et al., 2015*; *Honeyman et al., 2014*) (*Figure 1A*). We have recently shown that DNAJ-PKAc is solely expressed in FLCs, is cAMP-responsive, and importantly is incorporated into A-Kinase Anchoring Protein (AKAP) signaling complexes (*Riggle et al., 2016a*). This latter property provides a mechanism by which this pathological kinase is sequestered within defined subcellular locations and in immediate proximity to a subset of target substrates (*Langeberg and Scott, 2015*; *Scott and Pawson, 2009*; *Smith et al., 2017*).

While protein kinase A is generally not considered an oncogene, PKAc has been detected in the serum of patients with colon, renal, lungs, or adrenal carcinomas (*Cho et al., 2000*; *Cvijic et al., 2000*; *Porter et al., 2001*). Whole exome sequencing from independent patient cohorts have identified pathological mutations in PKAc that are linked to Cushing's syndrome (*Sato et al., 2014*). This disease occurs either as consequence of pituitary tumors that overproduce adrenocorticotropic hormone (ACTH) or as a consequence of aberrant signaling events that stimulate excess cortisol release from the adrenal glands (*Beuschlein et al., 2014*; *Lacroix et al., 2015*). In the latter instance, amino acid substitution of arginine 205 to lysine in PKAc prevents binding to the regulatory (R) subunits of PKA to promote mislocalization of uncontrolled PKA activity (*Cao et al., 2014*). In this report, we define a mechanism of action of DNAJ-PKAc, the fusion kinase exclusively expressed in fibrolamellar carcinoma. We have discovered that this fusion kinase is recruited into AKAP signaling complexes where, by virtue of its DNAJ domain, selectively interacts with the chaperonin heat shock protein 70 (Hsp70). This cellular chaperonin facilitates protein folding thereby providing an explanation as to why levels of DNAJ-PKAc protein are elevated over wildtype PKA in FLCs. The association of Hsp70 with DNAJ-PKAc also creates a unique therapeutic target for combinations of Hsp70 and kinase inhibitor drugs.

## Results

### DNAJ-PKAc in fibrolamellar carcinomas

Immunoblot screening of clinical samples with antibodies against PKAc revealed that human FLCs are heterozygous in that they express both wildtype PKA and the DNAJ-PKAc fusion (*Figure 1B*, top panel). This unique PKA fusion is solely expressed in FLCs, remains responsive to the second messenger cAMP, and importantly is incorporated by A-Kinase Anchoring proteins (AKAPs) into signaling complexes (*Riggle et al., 2016a*; *Riggle et al., 2016b*; *Turnham and Scott, 2016*). Immunofluorescent analysis of normal liver and FLC sections illuminated the distinctive morphology of this

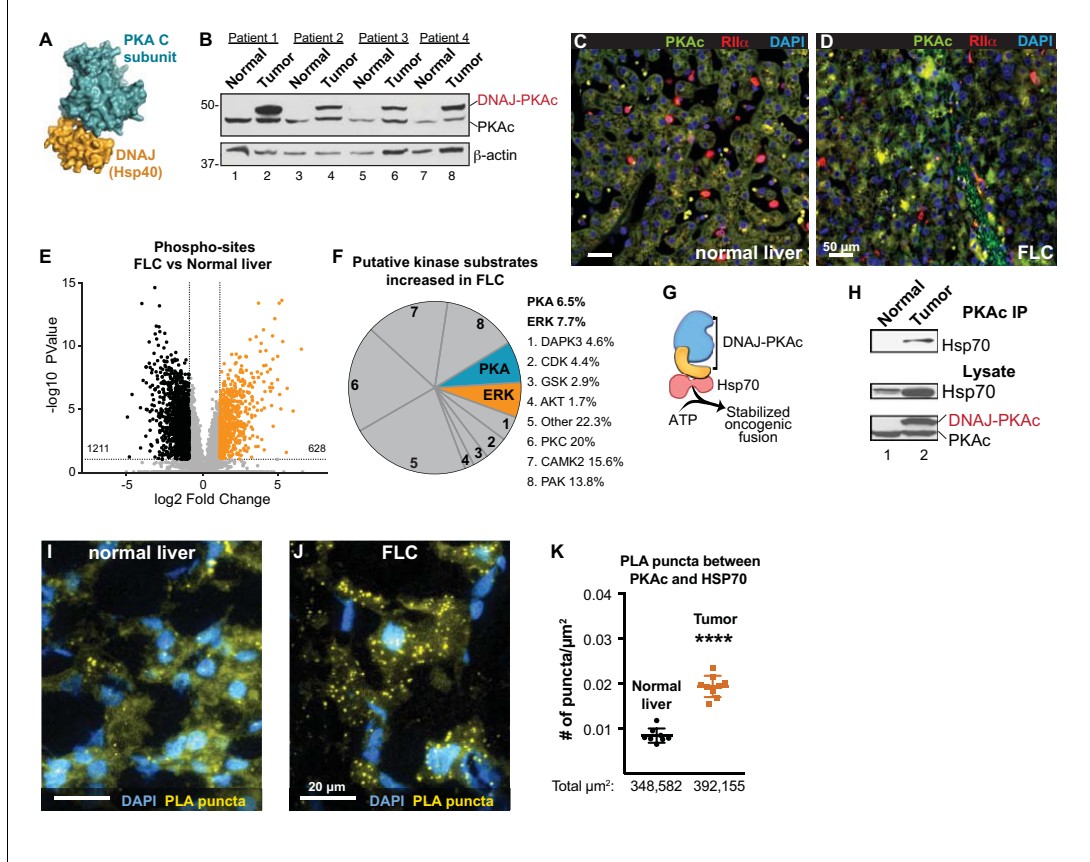

**Figure 1.** Properties of the DNAJ-PKAc fusion enzyme. (**A**) Structure of the DNAJ-PKAc fusion protein (PDB ID 4WB77). The DNAJ (orange) and PKAc domains (blue) are indicated. (**B**) Immunoblots of paired tumor and normal adjacent liver from FLC patients probed with antibodies to PKAc (top panels) and actin loading controls (bottom panels). DNAJ-PKAc (upper band) migrates with a slower mobility than the native C subunit in SDS-PAGE. (**C–D**) Immunofluorescence images of normal liver (left, **C**) and FLC (right, **D**) stained with antibodies against PKAc (green), RIIα (red) and DAPI (blue). Scale bar represents 50 μm. (**E–F**) Phosphoproteomic profiling of FLC. Statistical significance was calculated Significant differences in phosphopeptide expression between experiments were quantified with a two-tailed two sample t-test with unequal variances and Benjamini-Hochberg correction for multiple comparisons was applied (FDR ≤ 0.05), $\log_2$ ratio >1. (**E**) Volcano plot showing phosphosites upregulated (orange) and downregulated (black) in FLC as compared to normal adjacent liver. (**F**) Pie chart of putative kinase substrates (predicted by NetworKIN) increased in FLC. 82.8% of sites identified were in the NetworKIN platform. Percentages of sites ascribed to particular kinase are listed. 'Other' kinases include: CK1, TTK, GRK, RSK, MAK, JNK, ROCK, P70S6K, AMPK, CLK, HIPK2, PDHK, ACTR2, ATM, DMPK, IKK, MOK, NEK4, PKD1, PKG, TGFBR2, and p38-MAPK. (**G**) Schematic of DNAJ-PKAc in complex with heat shock protein 70 (Hsp70, red). (**H**) Immunoblot detection of Hsp70 in PKAc immune complexes from FLC and normal adjacent liver lysates (top). Loading controls indicate the levels of Hsp70 (middle) and both forms of PKA (bottom). (**I–J**) Proximity Ligation (PLA) detection of DNAJ-PKAc/Hsp70 complexes in (**I**) normal liver and J) FLC sections. Yellow puncta identify Hsp70-kinase sub-complexes, DAPI (blue) marks nuclei. Scale bar represents 20 μm. (**K**) Amalgamated data (PLA puncta/μm$^2$) from eight normal (black) and 9 FLC (orange) sections. Data are shown as mean ±s.d., p<0.0001 by Student's t-test (t = 10.98, df = 15).

DOI: https://doi.org/10.7554/eLife.44187.003

The following figure supplements are available for figure 1:

**Figure supplement 1.** Altered PKA signaling in FLC.

DOI: https://doi.org/10.7554/eLife.44187.004

**Figure supplement 2.** Kinase network rewiring in FLC.

DOI: https://doi.org/10.7554/eLife.44187.005

**Figure supplement 3.** Additional Proximity Ligation (PLA) detection of Hsp70 and PKAc in patient tissue.

DOI: https://doi.org/10.7554/eLife.44187.006

subtype of hepatocellular carcinoma where liver tumor is infiltrated with fibroid bands interspersed between cancerous hepatocytes (*Craig et al., 1980*). This 'intratumoral heterogeneity' is distinct from the undulating sinusoidal pattern of normal liver (*Figure 1C & D*). Co-localization of PKA catalytic (green) and regulatory subunits (RIIα, red) was evident in both sections. Counterstaining with DAPI (blue) is included to denote nuclei (*Figure 1C & D*). Additional biochemical characterization of these clinical samples substantiated the elevated expression of the type Iα regulatory subunit of PKA (RIα) in FLC tumors as compared normal adjacent tissue (*Figure 1—figure supplement 1A*, top panel) (*Riggle et al., 2016a*). Related experiments demonstrate that type II regulatory subunit (RII) levels do not fluctuate (*Figure 1—figure supplement 1A*, bottom panel).

The active site of DNAJ-PKAc is identical to that of the native kinase; both PKA forms are inhibited by PKI and are sensitive to the same spectrum of ATP analog inhibitors (*Cheung et al., 2015*; *Riggle et al., 2016a*). Immunoblot analyses using a phospho-PKA substrates antibody detects a different pattern of PKA phosphorylation in tumors as compared to adjacent liver extracts (*Figure 1—figure supplement 1B*). In addition, an RII overlay survey of AKAPs reveals a distinct pattern of anchoring proteins in FLC as compared to adjacent liver tissue (*Figure 1—figure supplement 1C*). These findings infer that introduction of DNAJ-PKAc results in changes in the substrate preference of this kinase or its access to subcellular targets. Yet, it remained important to ascertain whether the substrate specificity of this pathological fusion enzyme is altered in FLC. Phosphoproteomic profiling of human FLC and adjacent normal liver samples by label-free LC-MS/MS analysis identified 7697 phosphopeptides (*Hogrebe et al., 2018*) (*Figure 1E*; n = 6 technical replicates). Of these, 628 phosphopeptides were significantly enriched in FLCs as compared to adjacent normal liver (*Figure 1E*; orange). Substrate profiling with the NetworKIN platform predicted consensus kinase phosphorylation motifs (*Horn et al., 2014*). Of the phosphosites increased in FLC, 20% were putative PKC targets and 8% were ERK-MAPK sites (*Figure 1F*). This analysis revealed a systemwide rewiring of several protein kinase networks leading to increases and decreases in phosphorylation of specific substrates (*Figure 1—figure supplement 2*). Interestingly, PKA phosphosites were only enriched by 6.5%. However, phosphorylation of several key signaling effectors, scaffolding and anchoring proteins were enhanced (*Figure 1F* and *Figure 1—figure supplement 1D*). One plausible explanation for this surprisingly modest effect on PKA signaling is that oncogenesis driven by the fusion kinase may not only solely proceed through the kinase domain but also involves the chaperonin-binding site. Thus, DNAJ-PKAc may function to recruit additional elements that underlie the pathology of FLC (*Figure 1G*). Further immunoprecipitation experiments from clinical samples revealed that DNAJ-PKAc interacts with heat shock protein 70 (Hsp70; *Figure 1H*), a cellular chaperonin that facilitates protein folding and is frequently up-regulated in cancers (*Calderwood et al., 2006*; *Mayer and Bukau, 2005*). Proximity ligation (PLA) is an in situ technique that amplifies detection of native protein-protein interactions that occur within in a range of 40–60 nm (*Whiting et al., 2015*). This approach was used to identify interaction between endogenous Hsp70 and PKAc in liver sections from FLC patients (*Figure 1I,J & K*). PLA puncta indicative of native DNAJ-PKAc/Hsp70 sub-complexes were readily detected in regions of tumor (*Figure 1J* and *Figure 1—figure supplement 3*). In contrast, the number of PLA puncta was reduced in adjacent sections of healthy liver (*Figure 1I*). Quantification is presented in *Figure 1K and* additional PLA images of tissue sections are included in *Figure 1—figure supplement 3*. Recruitment of Hsp70 to DNAJ-PKAc may explain why protein levels of this fusion are frequently elevated compared to native PKA in FLCs (*Figure 1B*, top panel).

## Engineered disease-relevant AML12<sup>DNAJ-PKAc</sup> hepatocyte cell lines

FLC research to date has been hampered by the limited availability of patient samples, a paucity of disease-relevant cell-lines, and mouse models exhibiting a 24 month latency to develop hepatic tumors (*Engelholm et al., 2017*; *Kastenhuber et al., 2017*; *Oikawa et al., 2015*). Additionally, the most rigorously characterized PDX model is missing several key phenotypic traits of FLCs (*Oikawa et al., 2015*). Therefore, we employed CRISPR/Cas9 gene editing of chromosome eight in AML12 non-transformed murine hepatocytes to generate sustainable and homogenous cell lines. A 400 kb region was excised between intron 1 of the gene for Hsp40 (*Dnajb1*) and intron 1 of the gene for PKAc (*Prkaca*; *Figure 2A*). Initial characterization by PCR detected transcripts of intervening genes (*Gipc1*, *Ddx39* and *Lphn1*) located at the 5' end, middle and 3' end of the non-engineered strand of chromosome 8 (*Figure 2A & B*). Quantitative PCR measurement of mRNA transcripts for

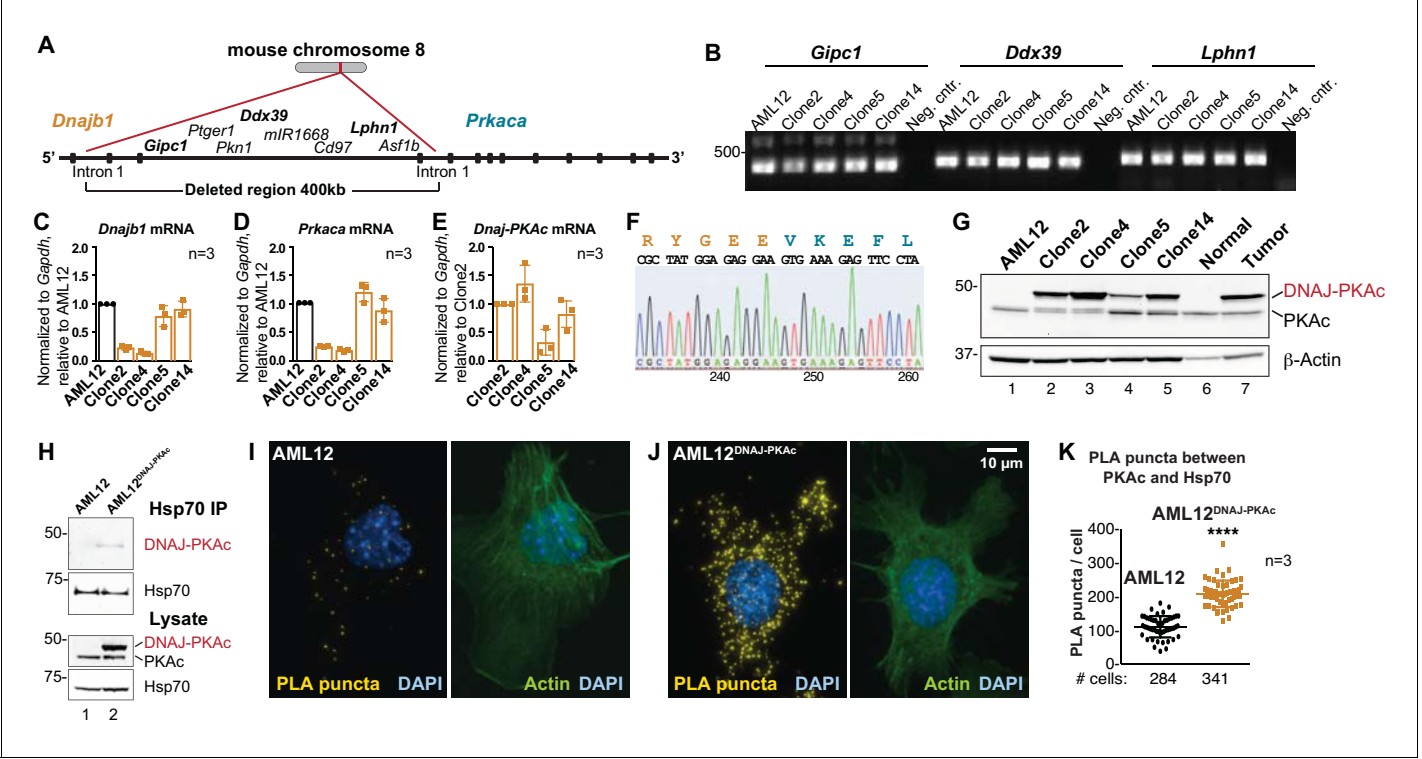

**Figure 2.** Generation and characterization of AML12^DNAJ-PKAc cell lines. (**A**) CRISPR-Cas9 gene editing of mouse chromosome eight in AML12 cells deleted a 400 kb region between intron 1 of the gene for Hsp40 (*Dnajb1*) and intron 1 of the gene for PKAc (*Prkaca*). (**B**) PCR detection of transcripts for the *Gipc1, Ddx39* and *Lphn1* genes encoded on the non-engineered strand of mouse chromosome 8. (**C–E**) Quantitative PCR detection of native mRNA transcripts in AML12 (black) and gene-edited (orange) cell lines. (**C**) Detection of native *Dnajb1* mRNA transcripts, (**D**) *Prkaca* transcripts and (**E**) *Dnajb1-Prkaca* mRNA transcripts. Data (n = 3) is normalized to *Gapdh* (C–E) and relative to (C,D) wildtype AML12 or (E) clone 2. Error bars indicate mean ±s.d. (**F**) Amino acid sequence of the fusion protein DNAJ-PKAc is shown in orange and blue. Nucleotide sequence of the fusion gene from clone 14 AML12^DNAJ-PKAc cells is shown below. (**G**) Immunoblot detection of both native and mutant PKAc in four clonal AML12^DNAJ-PKAc cell lines. Top) DNAJ-PKAc fusion proteins (upper bands) and wildtype PKAc (lower bands) are indicated. The distribution of PKAc in wildtype AML12 cells, normal liver and FLC are included. Bottom) Actin loading control. (**H**) Immunoblot detection of PKA in Hsp70 immune complexes isolated from wildtype (AML12) and clone 14 AML12^DNAJ-PKAc cells. Lysate loading controls indicate both forms of PKA (middle) and levels of Hsp70 (bottom). (**I and J**) Proximity Ligation (PLA) detection of proteins within 40–60 nm of each other in (**I**) AML12 and (**J**) AML12^DNAJ-PKAc cells. Yellow puncta identify Hsp70-kinase sub-complexes. Actin stain (green) marks cytoskeleton and DAPI staining (blue) marks nuclei. (**K**) Box-whisker plots of Hsp70-kinase sub-complexes. Amalgamated data (PLA puncta/cell) from AML12 (black) and AML12^DNAJ-PKAc (orange) cells. Number of cells analyzed over three independent experiments is indicated below each plot; data are shown as mean ±s.d., p<0.0001 by Student's t-test (t = 14.16, df = 105).
DOI: https://doi.org/10.7554/eLife.44187.007

The following figure supplements are available for figure 2:

**Figure supplement 1.** Additional characterization of AML12^DNAJ-PKAc cells.
DOI: https://doi.org/10.7554/eLife.44187.008

**Figure supplement 2.** Additional Proximity Ligation (PLA) detection of Hsp70 and PKAc in (**A**) AML12 and (**B**) AML12^DNAJ-PKAc cells.
DOI: https://doi.org/10.7554/eLife.44187.009

*Dnajb1* and *Prkaca* in wildtype and four gene-edited AML12^DNAJ-PKAc cell lines revealed differential expression of both transcripts in each clonal AML12^DNAJ-PKAc cell line (*Figure 2C & D*, **orange**). Likewise, the *Dnajb1-Prkaca* fusion transcript was present at different levels in each cell line (*Figure 2E*). Characterization by nucleotide sequencing and immunoblot analyses confirmed that these AML12^DNAJ-PKAc cell lines encode and express a single copy of DNAJ-PKAc (*Figure 2F & G*). As observed in FLCs, introduction of the DNAJ-PKAc allele promote the up-regulation of RIα expression (*Figure 2—figure supplement 1A*). Clone 14 was selected for further analyses as these cells express similar levels of DNAJ-PKAc and native PKA as compared to human FLC patients (*Figure 2G*). Interestingly, these clonal AML12^DNAJ-PKAc cells have similar levels of PKA activity and comparable migratory properties to the wildtype cell line (*Figure 2—figure supplement 1B–F*).

## Hsp70 is recruited to DNAJ-PKAc complexes

We next evaluated the formation of DNAJ-PKAc/Hsp70 complexes in our cell lines. Immunoblot analysis detected DNAJ-PKAc within Hsp70 immune complexes isolated from our AML12[DNAJ-PKAc] cell line, while PKAc was not present in Hsp70 immune complexes isolated from control AML12 cells (*Figure 2H, top panel*). Proximity ligation was used to evaluate DNAJ-PKAc/Hsp70 sub-complex formation (*Figure 2I & J*). In control cells, few puncta were evident when PLA was performed with antibodies against PKAc and Hsp70 (*Figure 2I*). In contrast, quantitation of PLA puncta (yellow) from >200 AML12[DNAJ-PKAc] cells revealed increased amounts of the DNAJ-PKAc/Hsp70 sub-complexes in our gene-edited cell lines (*Figure 2J & K*). Counterstaining with antibodies against actin (green) and DAPI (blue) defined whole-cell and nuclear boundaries, respectively. Additional PLA images from both cell types are included as *Figure 2—figure supplement 2*. Thus, our AML12[DNAJ-PKAc] cell line affords a disease relevant model with sufficient material to explore the mechanism of action of DNAJ-PKAc/Hsp70 assemblies.

Accelerated cell proliferation is a hallmark of carcinogenesis (*Hanahan and Weinberg, 2011*). Thus, three independent measurements assessed growth of AML12[DNAJ-PKAc] cells. First, cell proliferation was measured over 72 hr in culture using the MTS assay. Amalgamated data show that

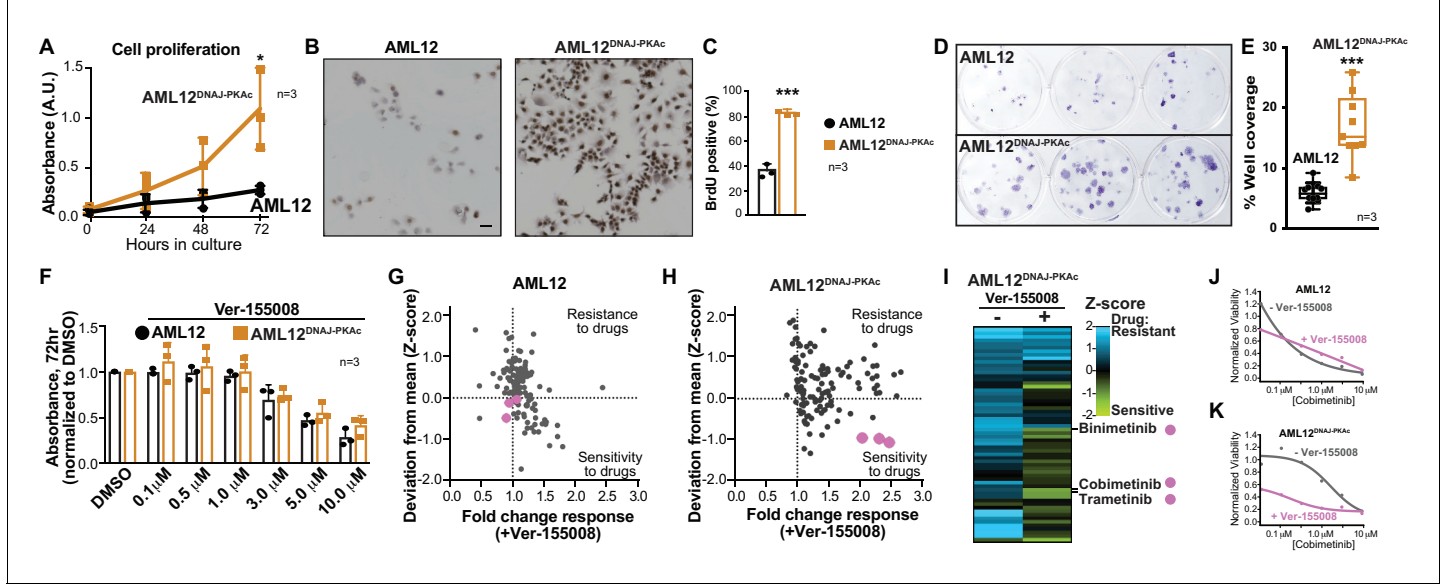

**Figure 3.** Cell proliferation analyses and combination drug sensitivity screening of AML12[DNAJ-PKAc] cells. (A) Cell growth of wildtype AML12 (black) and AML12[DNAJ-PKAc] (orange) cells measured by MTS colorimetric assay. Absorbance (AU) was measured over a time course of 72 hr. Data are expressed as mean ±s.d. (n = 3); p=0.01 (t = 4.49, df = 6). (B) In situ incorporation of BrdU as an independent means of assessing DNA synthesis. Representative panels of wildtype (left) AML12 and (right) AML12[DNAJ-PKAc] cells. Scale bar represents 50 μm. (C) Percentage of BrdU positive cells presented as mean ±s.d. (n = 3); p=0.0001 (t = 14.51, df = 4). (D) Clonogenic growth of (top) AML12 and (bottom) AML12[DNAJ-PKAc] cells. Cells were seeded at 200 cells/well in a 12 well plate and grown for two weeks in normal growth media followed by crystal violet staining. (E) Amalgamated data charting area of growth in each well is presented as box and whiskers plot (min-max; n = 3); p<0.0001 by Student's t-test (t = 6.14, df = 17). (F) Dose-response curves monitor the cytotoxic effects of the Hsp70 inhibitor Ver-155008 alone in AML12 (black) and AML12[DNAJ-PKAc] (orange) cells. Cell viability was assessed by MTS. Concentrations of drug used in each condition are indicated below each column. (G and H) Scatterplots show relative resistance or sensitivity of (G) AML12 and (H) AML12[DNAJ-PKAc] cells to the combination of 125 different chemotherapeutic drugs with Ver-155008. Drug combinations in the lower right quadrant are more sensitive to drug treatment than those in the upper right quadrant. Three drug combinations (pink circles) were identified for further validation, as they were more toxic to cells expressing DNAJ-PKAc than cells only expressing wildtype kinase. (I) Heat map of a subset of these data compares AML12[DNAJ-PKAc] cell survival with and without Ver-155008. AML12[DNAJ-PKAc] cells show drug resistance when treated with binimetinib, cobimetinib, or trametinib alone (left, blue) but they are more sensitive when these drugs are combined with Ver-155008 (right, green). (J and K) Analysis of (J) wildtype AML12 and (K) AML12[DNAJ-PKAc] cell survival. Dose-response of cobimetinib alone, (gray) or in combination with Ver-155008 (pink). Drug concentrations (μM) are indicated.

DOI: https://doi.org/10.7554/eLife.44187.010

The following figure supplement is available for figure 3:

**Figure supplement 1.** Repeat combination drug screens at lower concentrations (3 μM) of Ver-155008.

DOI: https://doi.org/10.7554/eLife.44187.011

AML12^DNAJ-PKAc cells proliferate more rapidly than wildtype AML12 cells (*Figure 3A*; *n = 3*). Second, immunostaining for BrdU incorporation showed that DNA synthesis is increased AML12^DNAJ-PKAc cells as compared to wildtype AML12 cells (82 ± 2% vs 36 ± 5%, *Figure 3B & C*; *n = 3*). Third, colony formation assays were performed to reinforce our data that AML12^DNAJ-PKAc cells have increased proliferative capacity as compared to their wildtype counterparts (*Figure 3D*; *n = 3*). Quantitation of amalgamated data confirms that AML12^DNAJ-PKAc cells proliferate more rapidly than their wildtype counterparts (*Figure 3E*). These findings lead us to surmise that the oncogenic nature of DNAJ-PKAc may not be simply due to changes in intrinsic kinase activity, but rather from the recruitment of Hsp70.

## Drug sensitivity screening in FLC model cells

A logical extension of this premise is to determine whether pharmacologically blocking Hsp70 influences proliferation of AML12^DNAJ-PKAc cells. Ver-155008 is an ATP-analog inhibitor of Hsp70 (IC50 = 0.5 µM) that halts cell proliferation in several cancer models (*Eugênio et al., 2017*; *Wen et al., 2014*). However, sole application of this drug over a range of concentrations did not have a differential effect on the viability of AML12^DNAJ-PKAc cells compared to wildtype AML12 cells as assessed by MTS assay at 72 hr (*Figure 3F*; n = 3). Consequently, we screened drug combinations that target additional elements within DNAJ-PKAc/Hsp70 signaling complexes. Cells were seeded and screened against a panel of 125 FDA-approved anti-cancer compounds in the presence or absence of Ver-155008 (*Pauli et al., 2017*) (*Figure 3G–K*). Cell viability was assessed by CellTiter-Glo assay and plotted against a standard deviation (Z-score) derived from collated mean responses (*Figure 3G & H*; *Pauli et al., 2017*; *Toyoshima et al., 2012*). Drug combinations in the lower right quadrant (*Sensitivity*) are more effective at reducing proliferation than drug combinations plotted in the upper right quadrant (*Resistance*). In wildtype AML12 cells, which lack the fusion enzyme, there was little change in the response to any of the FDA-approved drugs irrespective of whether the Hsp70 inhibitor was present (*Figure 3G*). Similarly, AML12^DNAJ-PKAc cells were refractory to most FDA-approved anti-cancer drugs in the absence of Ver-155008, but, when screening was repeated in the presence of Ver-155008 (over a range of concentrations up to 10 µM), certain drug combinations preferentially blunted AML12^DNAJ-PKAc cell proliferation (*Figure 3H & I*). Three Hsp70 inhibitor/drug combinations were appreciably more toxic to cells harboring DNAJ-PKAc than to cells only expressing wildtype kinase (*Figure 3G–I*, pink dots).

Deconvolution of our screening data revealed that these compounds were the MEK kinase inhibitors cobimetinib, binimetinib and trametinib. Further validation that these Hsp70/MEK inhibitor cocktails selectively target AML12^DNAJ-PKAc cells was obtained when the combination drug screen was repeated using lower doses of Ver-155008 (3 µM; *Figure 3—figure supplement 1*). Dose response curves revealed that wildtype AML12 cells are sensitive to cobimetinib alone (*Figure 3J*) whereas AML12^DNAJ-PKAc cells were more resistant to this drug over the same concentration range (*Figure 3K*). Importantly, in the presence of Ver-155008 the cytotoxic effect of cobimetinib in AML12^DNAJ-PKAc cells was enhanced (*Figure 3K*). Taken together, this screening venture provides two exciting new pieces of information: inhibition of Hsp70 in conjunction with blocking the RAF-MEK-ERK kinase cascade selectively affects the growth of cells expressing a single allele of DNAJ-PKAc, and drug combinations that target DNAJ-PKAc/Hsp70 assemblies offer a therapeutic strategy for FLC that warrants further investigation.

## Heterogeneous activation of the ERK signaling cascade in FLCs

A hallmark of FLC is the presence of fibroid bands that are interspersed between cancerous hepatocytes (*Craig et al., 1980*). This morphological feature is indicative of 'intratumoral heterogeneity' which promotes microenvironmental diversity in the primary liver cancer ecosystem (*Liu et al., 2018*; *Pribluda et al., 2015*). Through a combination of biochemical, imaging and proteomic approaches we show that intratumoral heterogeneity influences ERK signaling within FLCs. Immunoblot analyses of tumor lysates detect a slight reduction in global phospho-ERK signal in patient samples (*Figure 4A*, top panel). Yet immunofluorescent staining of tumor sections reveals clusters of prominent phospho-ERK signal in the cancerous hepatocytes (*Figure 4B & C*, yellow; *from patient 3*). Such regional detection of phospho-ERK is consistent with heterogeneous activation of the ERK cascade within the tumor. Likewise, the phosphoproteomic screen presented in *Figure 1E & F* identifies

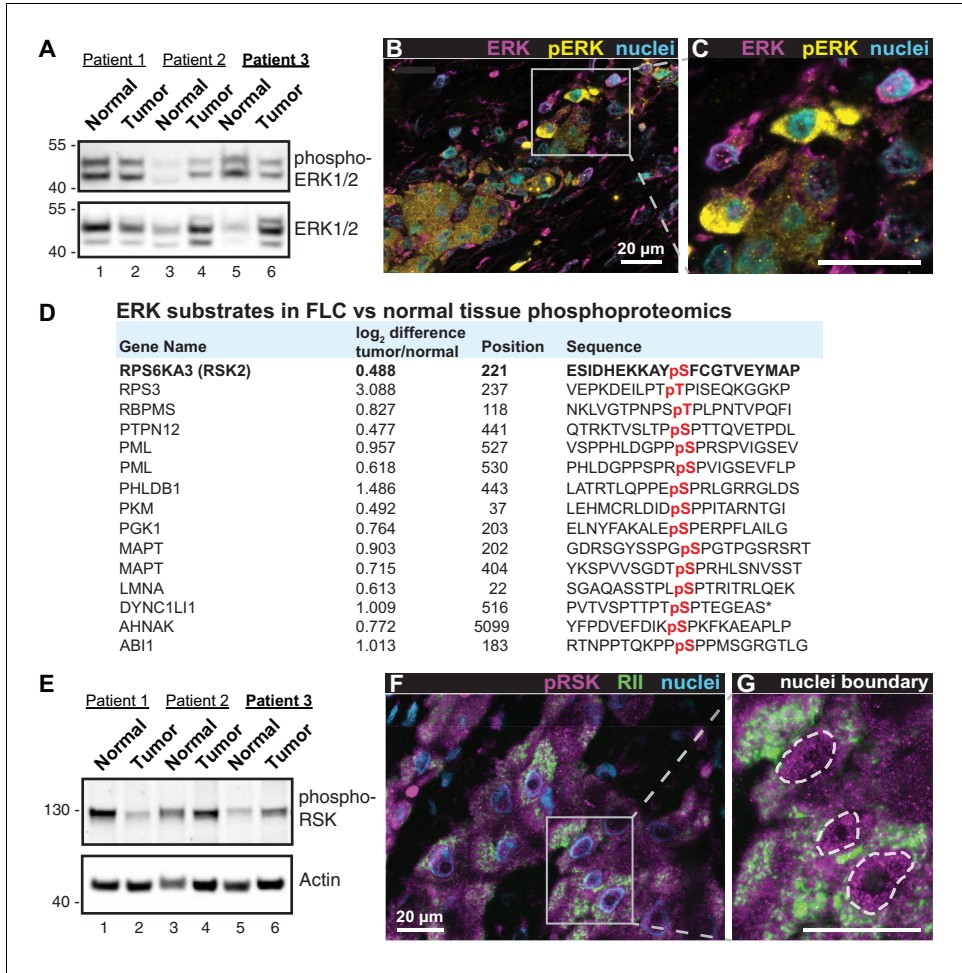

**Figure 4.** Heterogeneous activation of ERK signaling in FLCs. (**A**) Immunoblots of paired tumor and normal adjacent liver from FLC patients probed with antibodies to phospho-ERK1/2 (top panel) and total ERK1/2 (bottom panel). (**B**) Immunofluorescence images of FLC section from patient #3 were stained with antibodies against phospho-ERK (yellow), total ERK (magenta) and DAPI (*nuclei*, blue). Scale bar represents 20 μm. (**C**) Enlarged region from (**B**) showing prominent phospho-ERK staining in a subset of tumor hepatocytes. (**D**) Salient ERK substrates identified in phosphoproteomic analysis of FLC. Gene names, degree of enrichment (log$^2$difference tumor/normal) and primary phosphosite sequences (one letter code) are indicated. The protein kinase P90-RSK2 is highlighted. (**E**) Immunoblots of paired tumor and normal adjacent liver from FLC patients probed with antibodies to phospho-P90RSK (top panel). Actin loading control (bottom panel). (**F**) Immunofluorescence image of FLC section stained with antibodies against phospho-P90RSK (magenta), PKA RII (green) and the nuclear marker DAPI (blue). (**G**) Enlarged region from (**F**). Dashed lines) highlight increased nuclear accumulation of phospho-P90RSK signal. Scale bars indicate 20 μm.

DOI: https://doi.org/10.7554/eLife.44187.012

numerous ERK substrates that are elevated in FLC tumor as compared to normal liver (*Figure 4D*). This includes the protein kinase P90RSK, a well-characterized downstream target of ERK (*Dalby et al., 1998*). Validation of this ERK phosphorylation event is provided in two ways. First, immunoblot detection of pSer 221-P90RSK indicates variable activation of this kinase in the same cohort of FLC samples (*Figure 4E*, top panel). Second, immunofluorescent detection of phospho-P90RSK in tissue sections of FLC uncovered clusters of cells containing activated kinase (*Figure 4F & G*, magenta; from patient 3). Collectively these findings infer that the RAF-MEK-ERK kinase cascade is active in a subset of cells within the heterogeneous intratumoral environment of FLCs.

## AKAP-Lbc scaffolds promote ERK activation in FLC

On the basis of our understanding of how local signaling events are organized, we reasoned that AKAPs may be integral components of DNAJ-PKAc complexes (*Smith et al., 2017*). A logical candidate was AKAP-Lbc, a multifunctional anchoring protein and enhancer of ERK signaling (*Smith et al., 2010*) that interacts with another scaffolding protein, kinase suppressor of Ras (KSR), to form the core of a signaling network that integrates cAMP regulation of RAF-MEK-ERK signaling (*Figure 5A*). We found that AKAP-Lbc protein is up-regulated in human FLCs as compared to normal adjacent liver (*Figure 5B*, top panel, lane 2) and immunoblot analysis detected DNAJ-PKAc in AKAP-Lbc immune complexes isolated from FLCs (*Figure 5C*, top panel, lane 2). Parallel experiments show that DNAJ-PKAc/Hsp70 sub-complexes co-fractionate with this anchoring protein in AML12$^{DNAJ-PKAc}$ cells (*Figure 5D*, top panel, lane 2). Thus, AKAP-Lbc can sequester Hsp70 and DNAJ-PKAc with an ERK signaling module in AML12$^{DNAJ-PKAc}$ cells and human FLCs.

Detection of phospho-ERK1/2 is frequently used as a biochemical readout for activation of the RAF-MEK-ERK kinase cascade (*Rossomando et al., 1992*). Notably, basal levels of phospho-ERK 1/2 were elevated 2.8 ± 1.5 fold (n = 4) in AML12$^{DNAJ-PKAc}$ cells as compared to wildtype controls (*Figure 5E*). This finding was confirmed in situ by immunofluorescent detection. Phospho-ERK signal was barely detectable in control AML12 cells (*Figure 5F & G*), but clearly evident in the cytoplasm of AML12$^{DNAJ-PKAc}$ cells (*Figure 5H & I*). Actin (red) and DAPI (blue) were used as cytoskeletal and nuclear markers respectively (*Figure 5G & I*). We next monitored the efficacy of Hsp70/MEK inhibitors on basal ERK activity in AML12$^{DNAJ-PKAc}$ cells. In wildtype cells, treatment with Ver-155008 (3 µM) alone had no effect on ERK activation (*Figure 5J*, top panel, lanes 1 and 2). However, administration of cobimetinib (100 nM) or a combination of both drugs abolished detection of the phospho-ERK signal (*Figure 5J*, top panel, lanes 3 and 4). In contrast, basal phospho-ERK levels were high in AML12$^{DNAJ-PKAc}$ cells, treatment with Ver-155008 (3 µM) alone had a modest effect on phospho-ERK signal (*Figure 5J*, top panel, lanes 5 and 6). Application of cobimetinib (100 nM) or in combination with Ver-155008 abolished detection of phospho-ERK signals (*Figure 5J*, top panel, lanes 7 and 8). Thus, dual inhibition of Hsp70 and elements of the RAF-MEK-ERK cascade impedes mitogenic signals to preferentially block proliferation of AML12$^{DNAJ-PKAc}$ cells. This postulate was confirmed by clonogenic growth assays that monitor colony formation. Crystal violet staining showed that the synergistic effect of cobimetinib (100 nM) and Ver-155008 (3 µM) blocked AML12$^{DNAJ-PKAc}$ cell proliferation more potently than either drug alone (*Figure 5K*). Qualitatively similar results were obtained when parallel experiments were conducted with the more potent MEK inhibitor trametinib (30 nM) (*Figure 5—figure supplement 1*).

One intriguing outcome of our study is the question of whether or not interrupting the association between DNAJ-PKAc and Hsp70 impacts activation of the RAF-MEK-ERK cascade. Mutation of a conserved HPD motif that demarks a critical loop in the DNAJ domain abolishes interaction with Hsp70 (*Hennessy et al., 2005*) (*Figure 6A*). Thus, substitution of H33 to Q in the context of DNAJ-PKAc would be expected to prevent association with endogenous Hsp70 in AML12 cells (*Figure 6B*). Wildtype AML12 cells were transfected with vectors encoding DNAJ-PKAc or DNAJ-PKAc H33Q. Additional co-immunoprecipitation experiments used transiently transfected AKAP-Lbc as the scaffold to isolate DNAJ-PKAc-Hsp70 sub-complexes. Introduction of the H33Q mutation greatly reduces the level of Hsp70 in AKAP-Lbc complexes (*Figure 6C*, top panel, lane 3). The simplest explanation of this result is that addition of the J-domain onto the N-terminus of PKAc induces a novel interaction with Hsp70, thereby permitting the recruitment of this chaperonin to AKAP signaling islands. Immunoblot detection confirmed that basal levels of phospho-ERK were elevated upon introduction of DNAJ-PKAc in wildtype AML12 cells while transfection with the DNAJ-PKAc H33Q mutant diminished ERK activation (*Figure 6D*, top panel). Densitometry analysis of four independent experiments confirmed this result (*Figure 6D*, graph). Control immunoblotting monitored total ERK levels as a loading control and confirmed equivalent expression of each DNAJ-PKAc form in transfected cells (*Figure 6D*, lower panel).

Independent support for our hypothesis was provided through a phosphoproteomic screen that identified 2912 unique phosphopeptides in wildtype and AML12$^{DNAJ-PKAc}$ cells (*Figure 6E*). Of these, 96 phosphopeptides were increased (orange) and 76 were reduced in AML12$^{DNAJ-PKAc}$ cells (black). Substrate profiling using the NetworKIN platform revealed that 23% of ERK phosphosites were up-regulated in AML12$^{DNAJ-PKAc}$ cells whereas only 3% of PKA consensus sites were enriched

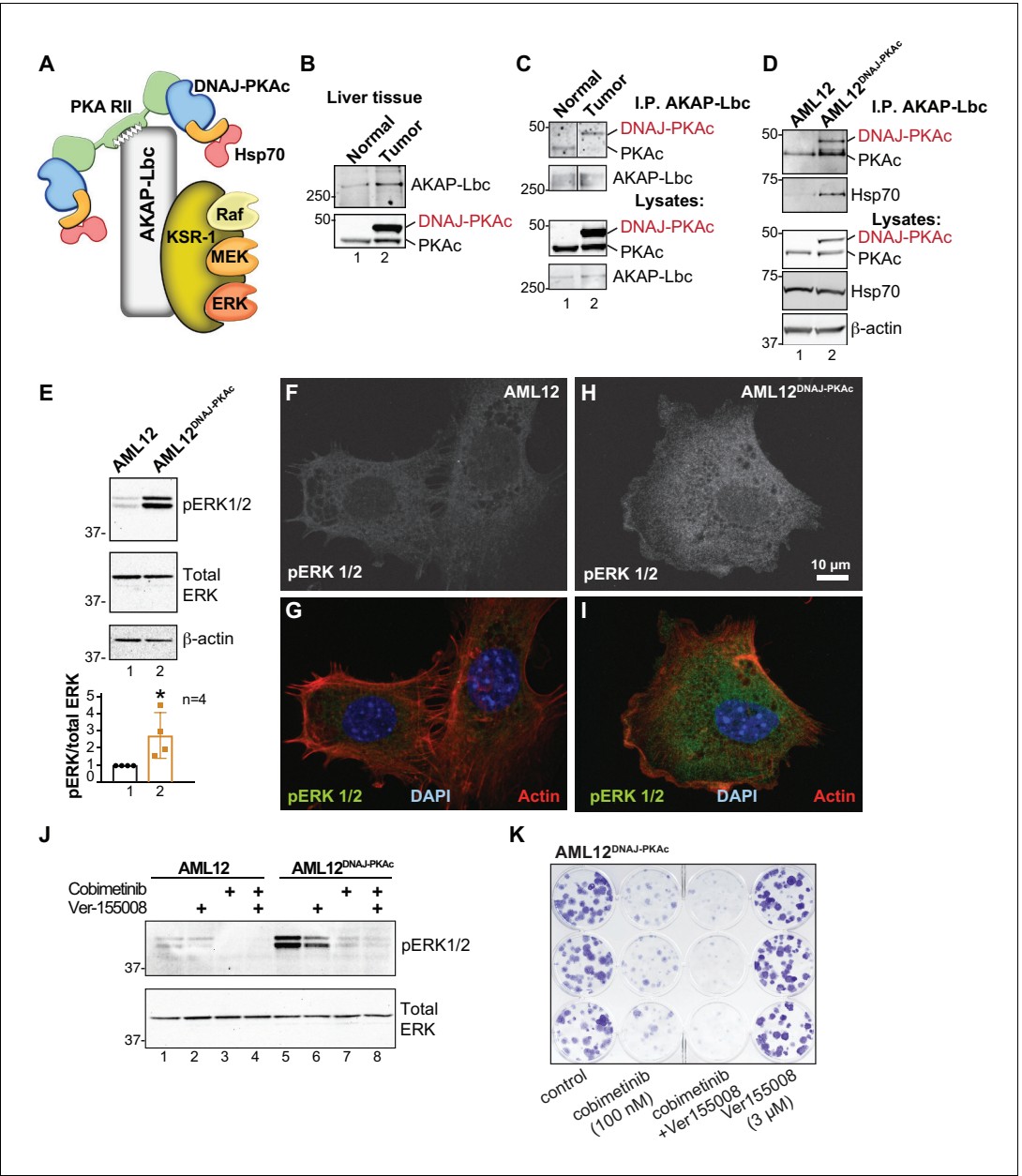

**Figure 5.** Pharmacologically targeting DNAJ-PKAc assemblies. (A) Schematic of an AKAP-Lbc-KSR-1 macromolecular assembly that sequesters Hsp70 and DNAJ-PKAc with elements of the ERK kinase cascade. (B) Immunoblots of paired FLC and normal adjacent liver probed with antibodies to AKAP-Lbc (top panels) and PKAc (bottom panels). (C) Immunoblot detection of PKAc (top) in AKAP-Lbc immune complexes (upper-mid) from normal adjacent tissue and FLC. PKAc (lower-mid) and AKAP-Lbc (bottom) in tissue lysates are indicated. DNAJ-PKAc (red) is indicated. (D) Co-immunoprecipitation of signaling elements with AKAP-Lbc from AML12$^{DNAJ-PKAc}$ cells. Immunoblot detection of PKAc (top) and Hsp70 (upper-mid) in immune complexes isolated from AML12$^{DNAJ-PKAc}$ cells. PKAc (middle), Hsp70 (mid-lower) in lysates from wildtype and AML12$^{DNAJ-PKAc}$ cells. Actin (bottom) served as loading control. (E) Immunoblot detection of phospho-ERK1/2 (top) as an index of ERK kinase activity in cell lysates from AML12 and AML12$^{DNAJ-PKAc}$ cells. Bottom) Immunoblot detection of total ERK served as a loading control. Quantification of immunoblots (n = 4); mean ±s.d. and p=0.04 (t = 2.6, df = 6). (f–I) In situ immunofluorescence of basal ERK activity. Grayscale images depicting immunofluorescent detection of phospho-ERK1/2 in (F) wildtype and (H) AML12$^{DNAJ-PKAc}$ cells. Composite images of phospho-ERK1/2 (green), actin (red) and nuclei (blue) in (G) wildtype and (I) AML12$^{DNAJ-PKAc}$ cells. Scale bar represents 10 μm. (J) Immunoblot detection of phospho-ERK 1/2 in wildtype AML12 (lanes 1–4) and AML12$^{DNAJ-PKAc}$ cells (lanes 5–8). Cells were treated with 100 nM of the MEK inhibitor cobimetinib, 3 μM Ver-155008 or combination of both drugs. Bottom)
*Figure 5 continued on next page*

*Figure 5 continued*

Detection of total ERK served as loading control. (K) Clonogenic growth assay portraying crystal violet (blue) staining of AML12$^{DNAJ-PKAc}$ cell proliferation in the presence of cobimetinib (100 nM), Ver-155008 (3 μM) and both drugs in combination.

DOI: https://doi.org/10.7554/eLife.44187.013

The following figure supplement is available for figure 5:

**Figure supplement 1.** Effect of combination treatment with trametinib and Ver-155008 on cell growth.

DOI: https://doi.org/10.7554/eLife.44187.014

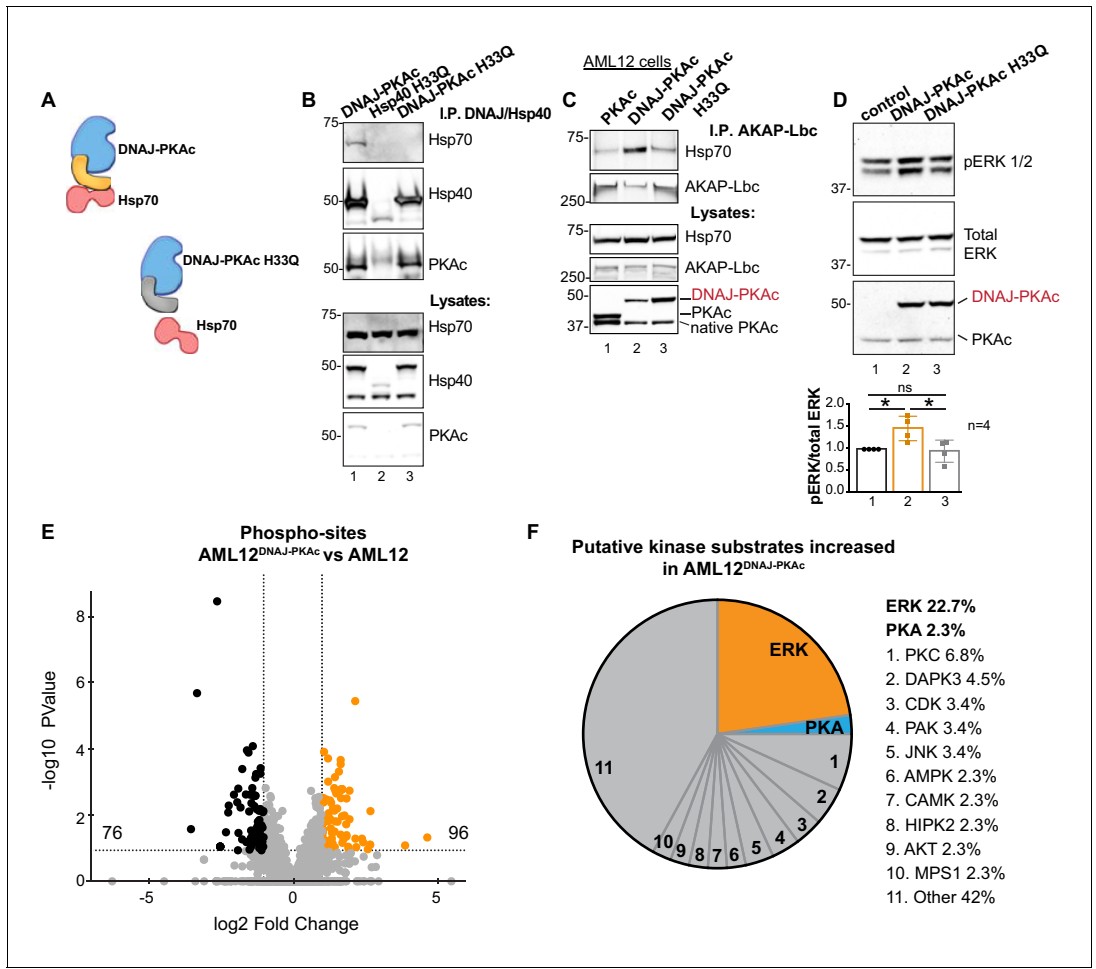

**Figure 6.** Interruption of the DNAJ-PKAc/Hsp70 interface reduces ERK activation: substrate bias towards ERK signaling in AML12$^{DNAJ-PKAc}$ cells. (A) Schematics of native DNAJ-PKAc (left) and DNAJ-PKAc H33Q mutant that cannot bind Hsp70 (right, gray). (B) Mutation of the chaperonin-binding site (H33Q) on DNAJ-PKAc abrogates interaction with Hsp70. Endogenous HSP70 co-precipitates with DNAJ-PKAc in AML12 cells expressing FLAG-DNAJ-PKAc (lane 1), but not with FLAG-Hsp40 H33Q control (lane 2) or the FLAG-DNAJ-PKAc H33Q mutant (lane 3). (C) GFP-tagged AKAP-Lbc co-precipitates endogenous Hsp70 in AML12 cells expressing FLAG-DNAJ-PKAc (lane 2) but not in cells expressing the wildtype FLAG-PKAc (lane 1) or the FLAG-DNAJ-PKAc H33Q mutant (lane 3). (D) Immunoblot detection of phospho-ERK1/2 in AML12 cells transiently transfected with DNAJ-PKAc (lane 2) or DNAJ-PKAc H33Q (lane 3). Total ERK (middle) served as a loading control. Detection of PKAc (bottom) monitored transfection efficiency. Quantitation of blots from four experiments, p=0.01 (t = 3.406, df = 6) and p=0.03 (t = 2.758, df = 6). (E and F) Differential phosphoproteomic profiling of AML12$^{DNAJ-PKAc}$ cells. (E) Volcano plot showing abundance (orange) and reduction (black) of phosphopeptides in AML12$^{DNAJ-PKAc}$ cells. Statistical significance of biological replicates was calculated by Student's t test with Log$_{10}$-transformed p-values of individual phosphopeptides plotted against log$_2$-transformed fold change; n = 6. (F) Pie chart of putative kinase substrates increased in AML12$^{DNAJ-PKAc}$ cells. Sites identified by NetworKIN platform. Individual kinases are listed. 'Other' kinases include: CK, ABL2, GRK, GSK3, JAK2, NLK, and SRC.

DOI: https://doi.org/10.7554/eLife.44187.015

(*Figure 6F*). Enrichment of PKC (7%), DAP kinase (5%) and CDK (3%) phosphosites were also evident. This systemwide analysis suggests that DNAJ-PKAc/Hsp70 macromolecular assemblies bias the signaling landscape toward ERK activation and mobilize other downstream kinase networks.

## Discussion

We have discovered that DNAJ-PKAc, a unique fusion protein that is emblematic of fibrolamellar carcinoma (FLC), functions as a scaffolding protein to assemble additional signaling elements that contribute to the pathogenesis of this cancer. More specifically, we show that the chaperonin-binding domain of this fusion enzyme supports recruitment of the co-chaperonin Hsp70. This creates a unique molecular context in which the DNAJ-PKAc chimera acts in FLCs. Chaperonopathies are a group of diseases caused by genetic lesions or aberrant post-translational modifications of molecular chaperones (*Macario and Conway de Macario, 2007*). 'Chaperonopathies by mistake' are a subgroup of related disorders, including certain cancers, in which chaperonin activity is normal, but becomes inappropriately assimilated into molecular pathways that enhance disease progression (*Macario and Conway de Macario, 2007*). We believe that formation of DNAJ-PKAc/Hsp70 subcomplexes in FLC is an example of this latter category (*Calderwood et al., 2006*; *Mayer and Bukau, 2005*; *Whitesell and Lindquist, 2005*). Chaperonins can repair misfolded proteins to reduce cellular stress or, as we believe is the case in FLC, recruitment of Hsp70 through the DNAJ domain preferentially stabilizes the chimeric PKAc fusion protein. This hypothesis is borne out by data in *Figures 1B* and *2G* wherein we demonstrate that protein levels of the DNAJ-PKAc variant are elevated in tumor samples and disease-relevant cell lines as compared to native PKA. In addition, the abnormal pairing of Hsp70 with DNAJ-PKAc creates new and unique drug targets. This rationale provided the impetus to screen a panel of recognized chemotherapeutics in combination with an Hsp70 inhibitor. This new precision pharmacology approach ascertained if certain drug pairings act synergistically to inhibit proliferation of cells harboring macromolecular complexes of this chaperonin and the fusion kinase.

Since aberrant kinase activity is known to drive many cancers, we further reasoned that augmented PKA activity could also contribute to the pathobiology of FLC (*Druker et al., 2001*; *Turnham and Scott, 2016*). However, one confounding factor is that the pathological DNAJ-PKAc fusion and its native kinase counterpart share similar sensitivities to the inhibitor PKI and efficiently bind R subunits to form type I and type II PKA holoenzymes (*Cao et al., 2019*; *Cheung et al., 2015*; *Riggle et al., 2016a*; *Scott et al., 1985*). Although physiochemically similar, notable differences between the PKA holoenzyme subtypes include lack of an autoregulatory phosphorylation site in RI isoforms, different in vitro binding affinities for cAMP and dispersal to distinct subcellular sites via interaction with distinct AKAPs (*Aggarwal et al., 2019*; *Burgers et al., 2012*; *Feramisco et al., 1980*; *Means et al., 2011*; *Smith et al., 2018*; *Taylor et al., 2012*). Another noteworthy feature is that expression of the DNAJ-PKAc enhances production or stabilization of the RIα subunit. Interestingly this phenomenon occurs in both FLCs and AML12$^{DNAJ-PKAc}$ cells (*Langeberg and Scott, 2015*; *Riggle et al., 2016a*) and *Figure 1—figure supplement 1A*). Such increased availability of RIα subunits may be indicative of tumor-specific variation in the ratio of type I to type II PKA activity. Switching of PKA isotypes may be clinically relevant as lesions in RI subunit genes are linked to disease (*Cho-Chung and Nesterova, 2005*; *Stratakis, 2013*). For example, nonsense and insertion mutations reduce levels of RIα in the endocrine neoplasia Carney complex (*Stratakis, 2013*). Similarly, mutations in the cAMP binding sites that render RIα less sensitive to cAMP have been linked to the rare skeletal dysplasia syndrome acrodysostosis type I (*Rhayem et al., 2015*). Yet, perhaps the most intriguing example is a single case report of inactivating mutations in RIα that induce sporadic fibrolamellar carcinomas in the absence of classic DNAJ-PKAc (*Graham et al., 2018*). Although the molecular mechanism surrounding this unusual case is not clear, one can postulate that reduced type I PKA activity, loss of anchoring to RI selective AKAPs, or overcompensation by type II PKA holoenzymes contributes to pathogenesis. Thus, marked changes in the quantity, isotype ratio and subcellular distribution of PKA holoenzymes, combined with the availability of DNAJ-PKAc may be factors that contribute to the etiology of FLC. A second postulate is that re-localization of Hsp70 to AKAP complexes by DNAJ-PKAc may be a critical event in transformation in FLC patients. Thus, the chaperonin binding properties of DNAJ-PKAc may as pertinent to oncogenesis as the intrinsic kinase activity of the fusion enzyme. Hence, we propose that the genetic lesion in chromosome 19 that is a

hallmark of FLC incorporates a new binding interface that transforms PKA from an essential 'homeostatic enzyme' into a dual-function kinase/scaffolding protein with pathological implications.

Our combination drug screen implicates mobilization of the ERK signaling cascade as a likely factor in the progression of FLC. Although mitogen-activated protein kinase (MAPK) pathways feature prominently in many cancers (*Kolch et al., 2015*), we propose that the impact of RAF-MEK-ERK signaling on FLC is complex and atypical. Three factors contribute to this view. First, whole exome sequencing confirms that FLCs lack activating mutations in Ras or B-RAF, but rather arise from a monogenic lesion in chromosome 19 that produces the DNAJ-PKAc fusion (*Cornella et al., 2015*; *Lalazar and Simon, 2018*; *Simon et al., 2015*; *Xu et al., 2015*). Second, our screen of FDA approved anti-cancer compounds in *Figure 3H and I* reveals that drugs targeting upstream elements of the ERK cascade, including EGF receptor antagonists erlotinib, lapatinib and afatinib and the B-RAF inhibitors dabrafenib and vemurafenib were ineffective, or at best exhibited modest anti-proliferative effects when used in combination with Ver-155008. Interestingly, the ERK inhibitor GDC-0994 had little effect on proliferation in these combination screens. Therefore, we interpret the exquisite sensitivity of AML12$^{DNAJ-PKAc}$ cells to MEK inhibition to suggest that DNAJ-PKAc may be acting downstream of Ras-Raf activation. A third contributing factor seems to be the atypical pattern of ERK activation in FLCs, which impacts downstream phosphorylation events. We base this conclusion on the regional immunofluorescent detection of phospho-ERK and its substrate P90RSK in FLC sections (*Figure 4C & G*). If these findings are reconciled with immunoblot data indicating that global levels of phospho-ERK and phospho-P90RSK minimally change tumor lysates, it argues for heterogeneous activation of both kinases occurs only in pockets of tumor. Collectively, these observations argue that the distinctive morphological features of FLC where cancerous cells are intermingled with fibrous tissue creates a heterogeneous tumor microenvironment that is prone to irregular activation of the ERK signaling cascade.

Although molecular links between ERK and DNAJ-PKAc were not immediately evident, we reasoned that one commonality was the proto-oncogene AKAP-Lbc. Anchored PKA activity has been implicated in the phosphorylation of RAF kinase and KSR-1 in the context of AKAP-Lbc signaling complexes (*Smith et al., 2017*; *Smith et al., 2010*; *Takahashi et al., 2017*). In addition, AKAP-Lbc is upregulated in human FLCs, and interacts directly with RAF-MEK-ERK kinase signaling scaffolds. In keeping with this molecular mechanism, our phosphoproteomic analysis identifies elevated PKA phosphorylation of serine 838 on KSR in FLCs (*Figure 1—figure supplement 1D*). This is especially interesting in light of early findings that Hsp90 and certain Hsp70 isoforms are elements of KSR scaffolds (*Stewart et al., 1999*) and data in *Figure 5B–D* showing that the chaperonins including the Hsp70-DNAJ-PKAc subcomplex are selectively recruited to AKAP-Lbc-KSR signaling units in FLC.

The relationship between cAMP and ERK signaling in cancer is complex and context dependent (*Dumaz and Marais, 2005*). For example, PKA has pleotropic effects on tumor-initiation. Paradoxically, a recent report postulates that PKA activity leads to mesenchymal-to-epithelial transitions that impede oncogenesis; yet DNAJ-PKAc kinase activity is thought to be necessary for tumor initiation (*Kastenhuber et al., 2017*; *Pattabiraman et al., 2016*). Therefore, one pertinent and unanswered question is whether or not the kinase activity residing within DNAJ-PKAc is an absolute requirement for FLC progression. This view is further substantiated by the phosphoproteomics data presented in *Figure 6E and F* showing that DNAJ-PKAc/Hsp70 macromolecular assemblies skew the signaling landscape toward enhanced ERK signaling rather than simply potentiating the action of PKA. Together, these results imply that recruitment of Hsp70 enhances basal ERK signaling in AML12$^{DNAJ-PKAc}$ cells by preferentially stabilizing this oncogenic signaling unit. Indirect support for this notion is presented in *Figure 6D* showing that abolishing the binding of Hsp70 to DNAJ-PKAc complex decreases ERK signaling. Thus, we postulate that the preferential stabilization of DNAJ-PKAc proceeds through the local action of Hsp70. Such a mechanism could explain why greater amounts of the DNAJ-PKAc fusion kinase are detected in FLCs and our AML12$^{DNAJ-PKAc}$ cells as compared to wildtype PKA. That said, we do not discount the kinase activity of DNAJ-PKAc as a pathological factor in FLC. Rather, we propose that recruitment of Hsp70 via the DNAJ domain in the chimeric DNAJ-PKAc kinase is an important new element that contributes to the dysregulation of this unique fusion enzyme.

One shared objective of the FLC research community and those investigating rare adolescent cancers is to identify and test therapeutic targets in the most efficient possible manner (*Kastenhuber et al., 2019*). One advantage of screening FDA-approved compounds is that the

pharmacotoxicity, therapeutic indices, and off-target effects of most components are well documented. Accordingly, each MEK inhibitor identified in our screen has been approved for the treatment of melanoma and other cancers (*Caunt et al., 2015*). Another benefit of the combination screening approach is the potential to identify drug pairings that can be used at lower effective doses; though there is also the possibility that new drug combinations may prove more toxic. This could be an important consideration for Ver-155008 as clinical trials with other Hsp70 inhibitors hold promise for the treatment of cancers (*Goloudina et al., 2012*). Although the utility of Hsp70 and MEK inhibition as combination therapy for FLC is far from clear, our discovery of drug pairs that selectively halt the growth of cells expressing DNAJ-PKAc but not wildtype hepatocytes provides a valuable tool to further the investigation for new treatments of this debilitating disease of adolescents.

# Materials and methods

## Key resources table

| Reagent type (species) or resource | Designation | Source or reference | Identifiers | Additional information |
|---|---|---|---|---|
| Antibody | ActinGreen-488 | Molecular probes | R37110 | Manufacturer instructions |
| Antibody | ActinRed-555 | Molecular probes | R37112 | Manufacturer instructions |
| Antibody | AKAP-Lbc (VO96) | *Diviani et al., 2001* | rabbit polyclonal | (1:1000) |
| Antibody | Amersham ECL Mouse IgG, HRP-linked F(ab')$_2$ fragment (from sheep) | GE Life Sciences | NA9310 | (1:10000) |
| Antibody | Amersham ECL Rabbit IgG, HRP-linked F(ab')$_2$ fragment (from donkey) | GE Life Sciences | NA9340 | (1:10000) |
| Antibody | Actin beta | Sigma-Aldrich | A1978 mouse monoclonal RRID:AB_476692 | (1:2500) |
| Antibody | BrdU | Dako | M0744 mouse monoclonal RRID:AB_10013660 | (1:1000) |
| Antibody | Donkey anti-Mouse IgG, Alexa Fluor 555 | Invitrogen | A-31570 | (1:500) |
| Antibody | Donkey anti-Mouse IgG, Alexa Fluor 488 | Invitrogen | A-21202 | (1:800) |
| Antibody | Donkey anti-Rabbit IgG, Alexa Fluor 488 | Invitrogen | R37118 | (1:500) |
| Antibody | Donkey anti-Rabbit IgG, Alexa Fluor 555 | Invitrogen | A-31572 | (1:800) |
| Antibody | GAPDH-HRP | Novus | NB110-40405 mouse monoclonal RRID:AB_669249 | (1:1000) |
| Antibody | Hsp70 | Proteintech | 10995–1 rabbit polyclonal RRID:AB_2264230 | WB (1:500), PLA in tissue (1:200), PLA in cells (1:500) |
| Antibody | p-44/42 ERK | CST | 9102 rabbit polyclonal RRID:AB_330744 | (1:1000) |

*Continued on next page*

Continued

| Reagent type (species) or resource | Designation | Source or reference | Identifiers | Additional information |
|---|---|---|---|---|
| Antibody | p-44/42 ERK | BD Transduction | 610123 mouse monoclonal RRID:AB_397529 | WB (1:1000), IHC (1:100) |
| Antibody | phospho-p44/42 MAPK | CST | 9101 rabbit polyclonal RRID:AB_331646 | WB (1:500), IHC (1:100) |
| Antibody | PKAc | BD Transduction | 610981 mouse monoclonal RRID:AB_398294 | WB (1:500), PLA in tissue (1:200), PLA in cells (1:500) |
| Antibody | PKAc | CST | 5842 rabbit monoclonal RRID:AB_10706172 | IHC (1:500) |
| Antibody | RIa | BD Transduction | 610610 mouse monoclonal RRID:AB_397944 | (1:1000) |
| Antibody | RIIa | BD Transduction | 612243 mouse monoclonal RRID:AB_399566 | (1:1000) |
| Antibody | RIIb | BD Transduction | 610626 mouse monoclonal RRID:AB_397958 | (1:1000) |
| Antibody | phospho-RSK | Thermo-Fisher | PA5-37829 rabbit polyclonal RRID:AB_2554437 | WB (1:500), IHC (1:100) |
| Antibody | FLAG M2 Magnetic Beads | Sigma-Aldrich | M8823 mouse monoclonal RRID:AB_2637089 | IP (1:40) |
| Antibody | GFP | Rockland | 600-101-215 goat polyclonal RRID:AB_218182 | WB (1:1000), IP (1:700) |
| Antibody | RI | BD Transduction | 610165 mouse monoclonal RRID:AB_397566 | (1:500) |
| Antibody | phospho-PKA substrates (RRXS*/T*) | CST | 9624 rabbit monoclonal RRID:AB_331817 | (1:1000) |
| Antibody | NeutrAvidin-HRP | Thermo-Fisher | 31030 | (1:5000) |
| Antibody | RIIa and b | *McCartney et al., 1995* | goat polyclonal | (1:200) |
| Cell line (*M. musculus*) | AML12 | ATCC | ATCC: CRL-2254 RRID:CVCL_0140 | Obtained from KJR by way of Nelson Fausto lab (original ATCC depositor) |
| Chemical compound, drug | DAPI | Thermo-Fisher | 62248 | Manufacturer instructions |
| Chemical compound, drug | ATP, [γ−32P]- 3000 Ci/mmol 10mCi/ml EasyTide, 100 μCi | Perkin-Elmer | BLU502A100UC | |
| Chemical compound, drug | BrdU | Invitrogen | B23151 | |
| Chemical compound, drug | Cobimetinib | Sigma-Aldrich | ADV465749767 | |
| Chemical compound, drug | Trametinib | Sigma-Aldrich | ADV465749287 | |

*Continued*

| Reagent type (species) or resource | Designation | Source or reference | Identifiers | Additional information |
|---|---|---|---|---|
| Chemical compound, drug | Dexamethasone | Sigma-Aldrich | D4902 | |
| Chemical compound, drug | DMEM/F-12 | Gibco | 11320033 | |
| Chemical compound, drug | Fetal Bovine Serum | Thermo-Fisher | A3382001 | |
| Chemical compound, drug | Gentamicin sulfate salt | Sigma-Aldrich | G1264 | |
| Chemical compound, drug | ITS Liquid Media Supplement | Sigma-Aldrich | I3146 | |
| Chemical compound, drug | Lipofectamine LTX with Plus Reagent | Thermo-Fisher | 15338100 | |
| Chemical compound, drug | Puromycin | Sigma-Aldrich | P8833 | |
| Chemical compound, drug | TransIT-LT1 Transfection Reagent | Mirus | MIR2300 | |
| Chemical compound, drug | Trypsin-EDTA (0.25%), phenol red | Gibco | 25200056 | |
| Chemical compound, drug | Crystal Violet | Sigma | C3886 | |
| Chemical compound, drug | Ver-155008 | Sigma-Aldrich | 1134156-31-2 | |
| Commercial assay or kit | CellTiter 96 AQueous One Solution Cell Proliferation Assay | Promega | G3582 | |
| Commercial assay or kit | CryoGrinder Kir | OPS Diagnostics | CG0801 | |
| Commercial assay or kit | Duolink In Situ Orange Starter Kit Mouse/Rabbit | Sigma-Aldrich | DUO92102 | |
| Commercial assay or kit | GeneJET Genomic DNA purification kit | Thermo | K0721 | |
| Commercial assay or kit | Pierce BCA Protein Assay Kit | Thermo | 23225 | |
| Commercial assay or kit | PowerUp SYBR Green Master Mix | Thermo-Fisher | A25741 | |
| Commercial assay or kit | Reverse Transcription Supermix | Bio-Rad | 1708840 | |
| Commercial assay or kit | RNeasy Mini Kit | Qiagen | 74106 | |
| Commercial assay or kit | SignaTECT cAMP-Dependent Protein Kinase (PKA) Assay System | Promega | V7480 | |
| Commercial assay or kit | Zero Blunt TOPO PCR Cloning Kit | Thermo-Fisher | 450245 | |
| Peptide, recombinant protein | RII-biotin | *Carr et al., 1992* | | |

*Continued on next page*

Continued

| Reagent type (species) or resource | Designation | Source or reference | Identifiers | Additional information |
|---|---|---|---|---|
| Peptide, recombinant protein | PKI | Sigma-Aldrich | P7739 | |
| Recombinant DNA reagent | DNAJ-PKAc FLAG | This paper | | In-house modified pDEST12.2 (N-terminal FLAG) |
| Recombinant DNA reagent | DNAJ-PKAc H33Q FLAG | This paper | | In-house modified pDEST12.2 (N-terminal FLAG) |
| Recombinant DNA reagent | DNAJB1 FLAG | This paper | This paper | In-house modified pDEST12.2 with N-terminal FLAG; backbone from Invitrogen (discontinued) |
| Recombinant DNA reagent | AKAP-Lbc GFP | Clonetech; *Diviani et al., 2001* | | pEGFP-N1 (Clontech) backbone |
| Recombinant DNA reagent | hSpCas9-gDnajb1 -Prkaca-2A-Puro | This paper | RRID: Addgene_48138 | PX458 backbone; Dual U6-sgRNA cassettes |
| Sequenced-based reagent | Gipc1_F | This paper | PCR primers | GGGAAAGGACA AAAGGAACCC |
| Sequenced-based reagent | Gipc1_R | This paper | PCR primers | CAGGGCATTTG CACCCCATGCC |
| Sequenced-based reagent | Ddx39_F | This paper | PCR primers | CCGGGACTTTC TACTGAAGCC |
| Sequenced-based reagent | Ddx39_R | This paper | PCR primers | GAATGGCCTG GGGAATACAC |
| Sequenced-based reagent | Lphn1_F | This paper | PCR primers | ACCCCTTCCAGA TGGAGAATGT |
| Sequenced-based reagent | Lphn1_R | This paper | PCR primers | TGGGCAAGCAT CTATGGCAC |
| Sequenced-based reagent | Dnajb1_ex2_F | This paper | qPCR primers | GGGACCAGA CCTCGAACAAC |
| Sequenced-based reagent | Dnajb1_ex2_R | This paper | qPCR primers | GGCTAATCCTG GCTGGATAGAT |
| Sequenced-based reagent | Prkaca_ex1_F | This paper | qPCR primers | AAGAAGGGCA GCGAGCAGGA |
| Sequenced-based reagent | Prkaca_ex1_R | This paper | qPCR primers | GCCGGTGCCA AGGGTCTTGAT |
| Sequenced-based reagent | Gapdh_F | This paper | qPCR primers | ATTTGGCCGT ATTGGGCGCCT |
| Sequenced-based reagent | Gapdh_R | This paper | qPCR primers | CCCGGCCTTC TCCATGGTGG |
| Sequenced-based reagent | Dnaj-PKAc_F | This paper | qPCR primers | ACGAGATCAAG CGAGCCTAC |
| Sequenced-based reagent | Dnaj-PKAc_R | This paper | qPCR primers | TTCCCACTCTC CTTGTGCTT |
| Software, algorithm | GraphPad Prism | GraphPad Prism (https://graphpad.com) | | |
| Software, algorithm | ImageJ | ImageJ (http://imagej.nih.gov/ij/) | | |

*Continued*

| Reagent type (species) or resource | Designation | Source or reference | Identifiers | Additional information |
|---|---|---|---|---|
| Software, algorithm | MaxQuant/ Andromeda | https://www. maxquant.org/ | | PMID: 19029910 |
| Software, algorithm | NetworKIN | http://networkin.info/ | | PMID: 24874572 |
| Software, algorithm | Perseus | https://maxquant. net/perseus/ | | PMID: 27348712 |
| Software, algorithm | PhosphoSitePlus | https://www. phosphosite.org | | |

## Human liver samples

Human FLCs with paired normal liver were consented for tissue donation under IRB-approved protocols (#31281 and #51710).

## Phosphoproteomics

Human FLC and normal adjacent liver was harvested according to above IRB and flash frozen. AML12 and AML12$^{DNAJ-PKAc}$ cells were grown on a 15 cm dish and after rinsing twice with ice cold PBS, cells were harvested in 750 μL of 6M aq. Guanidine hydrochloride (Gdn*HCl) containing 100 mM Tris, 5 mM TCEP*HCl, and 10 mM chloroacetamide (CAM), pH 8.5, using a cell scraper. Frozen human FLC specimens of ca. 100 mg wet weight were ground into a fine powder using the Cryo-Grinder Kit from OPS Diagnostics (Lebanon, NJ) and added to the Gdn*HCl buffer described above. Cell lysates were pipetted into 1.5 mL microtubes, voretexed briefly and heated to 95C for 5 min. Samples were then sonicated in a Qsonica cup sonicator (Newton, CT) at 100 W for 10 min (30 s on, 30 s off) on ice. Protein content was measured using the Pierce 660 nm assay reagent (Thermo Fisher Scientific, Waltham, MA). Aliquots of 300 μg of protein were pipetted into a new tube and diluted 2-fold with 100 mM triethylammonium bicarbonate (TEAB) pH = 8.5. 3 μg of sequencing-grade endoproteinase Lys-C (Wako, Richmond, VA) were added (1:100 ratio) and the mixture agitated on a thermomixer at 1400 rpm at 37°C for 2 hr. The mixture was diluted another 2-fold with 100 mM TEAB pH = 8.5 and 3 μg of trypsin were added. The mixture was agitated on a thermomixer at 1400 rpm at 37°C for overnight, acidified with formic acid (1% final), and cleared by centrifugation for 10 min at RT and 14,000 rcf. Peptides were extracted from the supernatant using Oasis HLB 1cc (10 mg) extraction cartridges (Waters, Milford, MA). Cartridges were activated by passing through 200 μL of methanol followed by 200 μL 80% aq. ACN containing 0.1% TFA, equilibrated with 400 μL 1% aq. formic acid. Peptides were loaded and then washed with 400 μL 1% aq. formic acid. Peptides were eluted with 300 μL 80% aq. ACN containing 0.1% TFA and directly subjected to the published batch IMAC phosphopeptide enrichment protocol with the following minor modifications (*Golkowski et al., 2016*; *Villén and Gygi, 2008*). 20 μL of a 50% IMAC bead slurry composed of 1/3 commercial PHOS-select iron affinity gel (Sigma Aldrich), 1/3 in-house made Fe3+-NTA superflow agarose and 1/3 in-house made Ga3+-NTA superflow agarose were used for phosphopeptide enrichment (*Ficarro et al., 2009*). The IMAC slurry was washed three times with 10 bed volumes of 80% aq. ACN containing 0.1% TFA and phosphopeptide enrichment was performed in the same buffer. Phosphopeptides were desalted using C18 StageTips according to the published protocol with the following minor modifications; after activation with 50 μL methanol and 50 μL 80% aq. ACN containing 0.1% TFA the StageTips were equilibrated with 50 μL 1% aq. formic acid. Then the peptides that were reconstituted in 50 μL 1% aq. formic acid were loaded and washed with 50 μL 1% aq. formic acid. The use of 1% formic acid instead of 5% aq. ACN containing 0.1% TFA prevents the loss of highly hydrophilic phosphopeptides.

## nanoLC-MS/MS phosphoproteomics analysis

The LC-MS/MS analyses were performed on a Thermo Fisher Scientific Orbitrap Elite instrument (AML12 cell lines) or a Thermo Fisher Scientific Orbitrap Fusion (human FLC specimens) as described previously with the following minor modifications (*Golkowski et al., 2017*). Peptide samples were

separated on a Thermo-Dionex RSLCNano UHPLC instrument (Sunnyvale, CA) using 20 cm long fused silica capillary columns (100 µm ID) packed with 3 µm 120 Å reversed phase C18 beads (Dr. Maisch, Ammerbuch, DE). For phosphopeptide samples the LC gradient was 120 min long with 3–30% B at 300 nL/min. LC solvent A was 0.1% aq.acetic acid and LC solvent B was 0.1% acetic acid, 99.9% acetonitrile. Data-dependent analysis was applied using Top15 selection with CID fragmentation.

## Computation of MS raw files

Raw files were analyzed by MaxQuant/Andromeda (*Olsen et al., 2010*) version 1.5.2.8 using protein, peptide and site FDRs of 0.01 and a score minimum of 40 for modified peptides, 0 for unmodified peptides; delta score minimum of 17 for modified peptides, 0 for unmodified peptides. MS/MS spectra were searched against the UniProt human database (updated July 22nd, 2015). MaxQuant search parameters: Variable modifications included Oxidation (M) and Phospho (S/T/Y). Carbamido-methyl (C) was a fixed modification. Max. missed cleavages was 2, enzyme was Trypsin/P and max. charge was 7. The MaxQuant 'match between runs' feature was enabled. The initial search tolerance for FTMS scans was 20 ppm and 0.5 Da for ITMS MS/MS scans.

## Data processing and statistical analysis

MaxQuant raw data were processed, statistically analyzed and clustered using the Perseus software package v1.5.6.095 (*Tyanova et al., 2016*). Human gene ontology (GO) terms (GOBP, GOCC and GOMF) were loaded from the Perseus Annotations file downloaded on the 01.08.2017. Expression columns (phosphopeptide MS intensities) were log2 transformed and normalized by subtracting the median log2 expression value of each column from each expression value of the corresponding column. Potential contaminant, reverse hits and proteins only identified by site were removed. Reproducibility was analyzed by column correlation (Pearson's r) and replicates that showed a variation of >0.25 in the r value compared to the mean r-values of all replicates of the same experiment were removed as outliers. Significant differences in phosphopeptide expression between experiments were quantified with a two-tailed two sample t-test with unequal variances and Benjamini-Hochberg correction for multiple comparisons was applied (FDR = 0.05).

## NetworKIN analyses

For human FLC and normal adjacent liver, significantly enriched phosphosites in FLC were input into the NetworKIN platform. For sites significantly enriched in AML12[DNAJ-PKAc] cells, the conserved phosphosite in human was identified in PhosphoSitePlus and then input into NetworKIN. Minimum score cutoff was 1.

## Cell lines and culture

AML12 mouse hepatocytes were used in this study. These cells were developed by the Nelson Fausto lab (*Wu et al., 1994*). The cells from this study came from Dr. KJR and are also available at ATCC (https://www.atcc.org/Products/All/CRL-2254.aspx). The cells were verified and mycoplasma free before beginning these studies and are currently being re-tested by STR and mycoplasma detection at IDEXX (Westbrook, ME). AML12 cells were cultured in DMEM/F12 supplemented with 10% FBS, 0.04 µg/mL dexamethasone, 0.1% gentamicin, 1 µg/mL recombinant human insulin, 0.55 µg/mL human transferrin, and 0.5 ng/mL sodium selenite. All cell lines were maintained in a 5% CO2 incubator at 37˚C. For lysates probed with phospho-ERK, cells were serum-starved for 16–24 hr and lysed. Serum-starved medium was prepared as above with the exception of addition of FBS. Cells for cobimetinib and Ver-155008 treatment were serum-starved for 16–24 hr and then incubated with 3 µM Ver-155008 for 30 min, and either DMSO or 100 nM Cobmetinib was incubated for 10 min. AML12 cells for *Figure 6B–D* were transfected with constructs as indicated with TransIT-LT1 (Mirus Bio). Cells for *Figure 6B & C* were collected for immunoprecipitation after 24 hr, while cells for *Figure 6D* cells were switched to serum-free media for 16–24 hr.

## Generation of CRISPR-edited AML12[DNAJ-PKAc] cells

Guide (g) RNAs were designed to target intron 1 of either mouse *Dnajb1* (GCATTCCGGGGATC TAGCGG) or *Prkaca* (GTAGTGCTGAGGAGAGTGGGG) in order to introduce DNA double-stranded

breaks in the regions similar to the deletion seen in FL-HCC. We engineered constructs expressing Cas9 and both guide (g)RNAs into SpCas9-2A-Puro V2.0 (Addgene plasmid number 62988) (*Ran et al., 2013*) and transfected the vector into AML12 cells using lipofectamine LTX with plus (Thermo Fisher) according to manufacturer's instructions. Cells were subjected to 2 µg/mL puromycin (Sigma) selection 48 hr post-transfection. After 3 days in puromycin-containing media, cells were clonally isolated. After selection, cells were dissociated using 0.25% trypsin-EDTA (Gibco) and 200 cells were plated into 15 cm$^2$ dish and incubated for 48–96 hr or until single-cell derived colonies were visible. Single-cell derived colonies were hand picked with cloning disks (3.2 mm diameter, Sigma-Aldrich) soaked with 0.25% trypsin-EDTA and plated into single wells of a 96-well plate. Genomic DNA was extracted (GeneJET Genomic DNA purification kit, Thermo Fisher) to screen clonally-isolated cells. Polymerase chain reaction (PCR) was performed to determine a heterozygous deletion. Primer sequences are found in the Key Resources Table.

## RNA and qPCR

Total RNA was extracted from wildtype AML12 and *Dnajb1-Prkaca* clones using trizol and RNeasy Mini Kits (Qiagen) and reverse transcribed using iScript Reverse Transcription Supermix for RT-qPCR (Bio-Rad) according to manufacturer's instructions. The cDNA was subjected to PCR with primers against *Dnajb1-Prkaca* fusion, and the resulting amplification was subjected to Sanger sequencing. Quantitative PCR was performed on ABI Fast 7500 using PowerUp SYBR Green Master Mix (Thermo Fisher) according to manufacturer's instructions with primers (see Key Resources Table) against *Dnajb1-Prkaca* fusion, wildtype *Dnajb1*, wildtype *Prkaca*. Data are reported as delta delta Ct after normalizing to *Gapdh*. *Dnaj-PKAc* cDNA was isolated from clone 14 and cloned into Zero Blunt TOPO PCR Kit (Thermo Fisher) and sequenced to verify the in-frame fusion.

## Immunoblotting

Cells and human FLCs were lysed in ice-cold RIPA buffer (10 mM Tris-HCl, 150 mM NaCl, 1% sodium deoxycholate, 1% Nonidet P-40, 0.1% SDS, 2 mM EDTA, 50 mM sodium fluoride) with protease inhibitors. Cleared lysate was measured using BCA Protein Assay (Pierce). Lysate was boiled in 1X LDS buffer (Thermo Fisher), separated on 4–12% NuPAGE gradient gels (Thermo Fisher) and transferred onto nitrocellulose using standard techniques. Membranes were incubated overnight at 4°C in 5% w/v milk with TBST and the following antibodies: PKAc (BD Transduction, 610981), Hsp70 (Proteintech, 10995–1), β-actin (Sigma-Aldrich, A1978), AKAP-Lbc (V096, 1 µg ml$^{-1}$), phospho p44/42 MAPK (CST, 9101), p44/42 (CST, 9102). Membranes were washed in TBST, incubated with HRP-labeled secondary antibodies (GE Life Sciences), washed as before and developed using ECL (Thermo Fisher) on an iBright FL1000. For re-probing, membranes were striped with 1X Re-Blot Plus Strong (Millipore) for 15 min and then re-blocked in Blotto before incubation with primary antibodies again. Densitometry for blot quantification was done using ImageJ software (NIH; http://rsb.info.nih.gov/ij).

## Immunoprecipitation

Human tissue and cell lysates were lysed in 0.5% or 1% Triton-X buffer (50 mM Tris-HCl, 130 mM NaCl, 20 mM NaF, 2 mM EDTA, 0.5% or 1% Triton-X with protease inhibitors). Lysates were precleared with IgG and protein A/G agarose beads (Millipore) then incubated with anti-PKAc, anti-HSP70, anti-GFP, or anti-AKAP-Lbc antibodies overnight at 4°C. Immunocomplexes were separated by incubation with protein A/G agarose beads for 2 hr at 4C and washed 4 × 1 mL in lysis buffer. For FLAG immunoprecipitation, lysates were incubated with anti-FLAG M2 magnetic beads (Sigma M8823) overnight. Immunocomplexes were washed 4 × 1 mL in lysis buffer.

## Migration and invasion

AML12 and AML12$^{DNAJ-PKAc}$ cells were plated on a 96-well plate and subjected to IncuCyte ZOOM 96-Well Scratch Wound Cell Migration and Invasion assay (Essen Bioscience). Matrigel (Corning) was used in invasion assays. Data are representative images of n = 3. Images collected every 45 min for 24 hr (migration assay) or 48 hr (invasion assay).

## Protein kinase A activity assay

SignaTECT cAMP-dependent Protein Kinase (PKA) Assay System (Promega, V7480) was used to measure kinase activity. Cells were lysed and PKA activity was measured according to protocol (ATP, [γ−32P]- 3000 Ci/mmol 10mCi/ml EasyTide; Perkin Elmer, BLU502A001MC). Experiments were carried out ±25 μM cAMP to stimulate PKA,±Kemptide substrate for normalization, and ±50 μM PKI to inhibit PKA.

## Immunofluorescence and proximity ligation assay (PLA)

AML12 cells were grown on coverslips and fixed with 4% paraformaldehyde/PBS for 20 min. After several washes in PBS, samples were permeabilized in 0.5% Triton X-100/PBS for 10 min and washed extensively in PBS. Cells were then subjected to PLA or immunofluorescence. Human liver tissue for PLA was fresh frozen, cut on a cryostat at 8 μm, and fixed in 4% paraformaldehyde/PBS at RT for 4 min. For PLA, samples were processed according to manufacturer's instructions with anti-mouse and anti-rabbit reagents (Sigma) using PKAc (BD Transduction, 610981) and Hsp70 (Proteintech, 10995–1). Z-stacks of fluorescent images were collected using a Keyence BZ-X710 using relevant filter cubes. Maximum intensity projections were quantified for puncta number using Fiji/ImageJ. For AML12 cell PLA, images were smoothened and a duplicate image was created for use as a mask. The duplicate file was thresholded to capture as many puncta as possible without significant blending of densely packed signal. The binary mask was then used to measure selected regions from the original image. Total cell number per field of view was counted as DAPI-stained nuclei. For quantification of human liver tissue PLA, unfocused light was removed using the Keyence haze reduction function. Puncta number and fluorescence intensity were measured by automation using Keyence hybrid cell counter set to detect thresholded puncta between 0 and 1.0 μm in diameter. Puncta counts were normalized to the total image area.

Human liver tissue for immunostaining was formalin fixed and paraffin embedded. Samples were permeabilized in 0.5% Triton X-100/PBS for 10 min. Images for immunofluorescence were immunostained with primary antibodies [PKAc (CST, 5842), PKA RIIα (BD Transduction, 610626), Erk (BD 610123), (phospho p44/42 MAPK (CST, 9101), or phospho P90RSK (Thermo Fisher PA5-37829)] overnight at 4C. Cells were washed three times in PBS and incubated with Alexa Fluor conjugated secondary antibodies (Thermo Fisher) for 2 hr at room temperature. Nuclei were stained with DAPI and samples were mounted on glass slides using ProLong anti-fade media (Invitrogen) or Aqua-Mount (Thermo Scientific). Images were taken on a Leica DMI6000B inverted microscope with a spinning disk confocal head (Yokagawa) and a CoolSnap HQ camera (Photometrics) controlled by MetaMorph 7.6.4 (Molecular Devices), or a BZ-X710 microscope (Keyence).

## MTS assay

Cells were seeded at 3,000 cells/well into 96-well plates, allowed to recover for 16–24 hr and either treated with Ver-155008, DMSO, or left untreated. MTS reagent (CellTiter 96 Aqueous One Solution, Promega) was added per the manufacturer's instructions, and absorbance was read at 490 nm 3 hr later.

## BrdU labeling

Wildtype AML12 and AML12$^{DNAJ-PKAc}$ cells were seeded at 20,000 cells/well on a 2-well chamber slide (Lab-Tek). Fourty-eight hours after plating, cells were treated with 25 μg/mL BrdU (Roche Diagnostics) for 4 hr. Cells were washed twice in ice-cold PBS and fixed with 100% ice-cold methanol. BrdU labeling was then determined by immunohistochemistry by using anti-BrdU antibody (DAKO).

## Colony growth

For clonogenic growth assays, cells were seeded at 200 cells/well in 12-well dishes. For inhibitor tests, drug was added following day to appropriate concentrations (100 nM cobimetinib or 30 nM trametinib; 3 μM Ver-155008) in normal growth media. Media/drug was refreshed every 5 days. After two weeks, cells were washed in PBS and fixed for 20 min in 4% paraformaldehyde/PBS. Cells were then stained with 0.1% crystal violet in 10% methanol, washed 3x with water and air dried for image capture. Images were quantified in ImageJ using masking and particle analysis to determine well surface area covered by stained cells. Data were further analyzed and plotted in Prism 7.

## Drug screen

Drug screening of AML12 and AML12$^{DNAJ-PKAc}$ cells was performed using a drug library assembled by SEngine Precision Medicine (Seattle, Washington) that includes FDA-approved chemotherapies as well as drugs in clinical development. The drug screens used a dilution series of the inhibitors that started at 10 µM and decreased in half-log units to a final concentration of ~41 nM. Initial combination screens were performed with 10 µM Ver-155008, a concentration well above the IC50 (in vitro IC50 0.5 µM-2.6 µM) to assure strong Hsp70 inhibition. Cells were tested in 2D and data evaluated as described (*Pauli et al., 2017*).

## Statistical analyses

Statistically significant differences between samples were calculated as indicated in figure legends, using Student's two-tailed t-test or ANOVA with post-hoc multiple comparisons for groups of 3 or more. All results are presented as the mean ±s.d unless otherwise indicated. Sample size (n) indicated the number of independent experiments represented in amalgamated data; total cell numbers used in experiments are indicated. P values of < 0.05 were considered statistically significant.

## Data availability statement

Raw mass spectrometry data has been uploaded to MassIVE, an NIH supported MS data repository (MSV000083167).

## Acknowledgements

We thank Katherine Forbush for technical support, Melanie Milnes and Jennifer Nelson for administrative support, and all members of the Scott Lab for critical discussions. Research reported in this publication was supported in part by NIH/NIDDK under award number R01DK119192 (JDS); support from the Fibrolamellar Cancer Foundation (FCF; JDS, RSY); NIH/NCI award R21CA201867 and a St. Baldrick's Foundation Research Award (KSR); NIH/NCI award R21CA177402 and NIH/OD award S10 OD021502 (S-EO); NIH 2T32CA080416 (RET); and NIH/NIDDK F32DK121415 (MHO). This work used an EASY-nLC1200 UHPLC and Thermo Scientific Orbitrap Fusion Lumos Tribrid mass spectrometer purchased with funding from a National Institutes of Health SIG grant S10OD021502 (S-EO).

## Additional information

### Funding

| Funder | Grant reference number | Author |
| --- | --- | --- |
| National Institutes of Health | 2T32CA080416 | Rigney E Turnham |
| National Institute of Diabetes and Digestive and Kidney Diseases | F32DK121415 | Mitchell H Omar |
| National Cancer Institute | R21CA177402 | Shao-En Ong |
| NIH Office of the Director | S10 OD021502 | Shao-En Ong |
| National Cancer Institute | R21CA201867 | Kimberly J Riehle |
| St. Baldrick's Foundation | Research Award | Kimberly J Riehle |
| Fibrolamellar Cancer Foundation | | Raymond S Yeung John D Scott |
| National Institute of Diabetes and Digestive and Kidney Diseases | R01DK119192 | John D Scott |

The funders had no role in study design, data collection and interpretation, or the decision to submit the work for publication.

## Author contributions

Rigney E Turnham, Conceptualization, Resources, Data curation, Formal analysis, Supervision, Validation, Investigation, Visualization, Methodology, Writing—original draft, Project administration, Writing—review and editing; F Donelson Smith, Heidi L Kenerson, Conceptualization, Data curation, Formal analysis, Supervision, Validation, Investigation, Visualization, Methodology, Writing—review and editing; Mitchell H Omar, Data curation, Formal analysis, Validation, Investigation, Visualization, Methodology, Writing—review and editing; Martin Golkowski, Data curation, Formal analysis, Investigation; Irvin Garcia, Data curation, Formal analysis; Renay Bauer, Formal analysis; Ho-Tak Lau, Data curation, Formal analysis, Validation, Investigation; Kevin M Sullivan, Conceptualization, Formal analysis, Funding acquisition, Validation, Investigation, Writing—original draft, Project administration, Writing—review and editing; Lorene K Langeberg, Conceptualization, Data curation, Formal analysis, Supervision, Funding acquisition, Writing—original draft, Project administration, Writing—review and editing; Shao-En Ong, Conceptualization, Resources, Data curation, Formal analysis, Supervision, Investigation, Project administration, Writing—review and editing; Kimberly J Riehle, Conceptualization, Resources, Formal analysis, Supervision, Funding acquisition, Investigation, Project administration, Writing—review and editing; Raymond S Yeung, Conceptualization, Data curation, Formal analysis, Supervision, Funding acquisition, Validation, Investigation, Visualization, Methodology, Project administration, Writing—review and editing; John D Scott, Conceptualization, Formal analysis, Supervision, Funding acquisition, Methodology, Writing—original draft, Project administration, Writing—review and editing

## Author ORCIDs

F Donelson Smith (iD) https://orcid.org/0000-0002-8080-7589
Lorene K Langeberg (iD) http://orcid.org/0000-0002-3760-7813
John D Scott (iD) https://orcid.org/0000-0002-0367-8146

## Decision letter and Author response

Decision letter https://doi.org/10.7554/eLife.44187.022
Author response https://doi.org/10.7554/eLife.44187.023

## Additional files

### Supplementary files

• Supplementary file 1. Combination drug screen data.
DOI: https://doi.org/10.7554/eLife.44187.016
• Supplementary file 2. Phosphoproteomic data from FLCs and AML12$^{DNAJ-PKAc}$ cells.
DOI: https://doi.org/10.7554/eLife.44187.017
• Transparent reporting form
DOI: https://doi.org/10.7554/eLife.44187.018

### Data availability

Raw mass spectrometry data has been uploaded to MassIVE, an NIH supported MS data repository (MSV000083167).

The following dataset was generated:

| Author(s) | Year | Dataset title | Dataset URL | Database and Identifier |
|---|---|---|---|---|
| Golkowski M, Turnham RT, Ong SE, Scott JD | 2017 | An acquired scaffolding function of the DNAJ-PKAc fusion enhances oncogenesis in Fibrolamellar carcinoma | http://doi.org/10.25345/C5F01X | MassIVE Repository, 10.25345/C5F01X |

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
