## [Decision Letter]

Thank you for submitting your article "An acquired scaffolding function of the DNAJ-PKAc fusion contributes to oncogenic signaling in fibrolamellar carcinoma" for consideration by *eLife*. Your article has been reviewed by three peer reviewers, one of whom is a member of our Board of Reviewing Editors, and the evaluation has been overseen by Jonathan Cooper as the Senior Editor. The reviewers have opted to remain anonymous.

The reviewers have discussed the reviews with one another and the Reviewing Editor has drafted this decision to help you prepare a revised submission.

Summary:

This study establishes an elegant in vitro model of FLC through genetically engineered creation of a DNAJ-PKAc fusion protein that is a potential contributor to the pathogenesis of this cancer. Evaluation of this model reveals an interaction of the DNAJ-PKAc fusion protein with HSP70 and difference in phosphorylation in substrates for multiple kinases in cells that express the fusion kinase. The authors focus on ERK signaling due to the selective effect of MEK inhibitors to synergize with the Hsp70 inhibitor Ver-155008. This is a strength of the study. However, there are also weaknesses. The study would be improved by the inclusion of some in vivo analysis and application to cancer therapy – for example, no studies of drug efficacy in vivo are presented and there are no PDX studies. However, this is mitigated by the focus of the present study on molecular mechanism – accordingly, this evaluation will focus on the mechanistic aspects of the study. It is noted that the data presented do not fully support the mechanistic conclusions that are presented.

Essential revisions:

1) Much of the Discussion section appears to be an extension of the Results section. Several figures are presented for the first time in this section. This should be corrected with all new data presented in the Results section of the manuscript.

2) The study is focused on ERK. Do FLCs show evidence of increased ERK activity?

3) Are AML12 DNAJ-PKAc cells transformed? The observation that 15/15 FLC samples contained the fusion protein (Honeyman et al., 2014) support its role in tumor formation. It is not clear if this fusion is a driver of tumorigenesis in the model the authors created, yet the authors refer to it as oncogenic (Abstract, Introduction, subsection “Hsp70 is recruited to DNAJ-PKAc complexes”). This point is important because a major conclusion of this study is that combination therapy with Ver-155008 plus MEK inhibitor may suppress FLC. If the AML12 DNAJ-PKAc cells are not transformed, the relevance of the Ver-155008 plus MEK inhibitor combination is diminished. Although in vivo experimentation is not deemed essential for the response to this critique, it should be noted that the efficacy of the proposed drug combination using a FCL model (e.g., FCL PDX, Oikawa et al., 2015) could be tested.

4) Were the three hits from the screen the only MEK inhibitors in the screen? Were ERK or Raf inhibitors included? If so, why do the authors think they were not effective? If not, do these inhibitors provide the same level of sensitivity as the MEK inhibitors when tested in combination with Ver-155008?

5) Figure 1 shows colocalization of PKAc and RIIa. While there is some overlap in FLC there is no evidence to suggest that this colocalization is in association with the DNAJ-PKAc fusion protein versus the native PKAc since all tumors appear to retain a native PKAc. Moreover, there is no insight into the relevance of RIIa versus other regulatory subunits, including other disease associated isoforms like RIa.

6) Figure 3. The characterization of PKA activity in AML12DNAJ-PKAc lacks a statistical comparison with the parental line. Furthermore, it is unclear if the level of PKA activity in AML12DNAJ-PKAc accurately reflects that of patients (documented in Figure 1). It may be best not to minimize the importance of the PKA activity of DNAJ-PKAc based on data in the AML12DNAJ-PKAc cells without first confirming that this model reflects PKA activity in FLC.

7) The proposed mechanism involves the interaction of DNAJ-PKAc with AKAP-Lbc (Figure 4). This conclusion is speculative because no studies of AKAP-Lbc function are presented. Specifically, does loss of AKAP-Lbc decrease DNAJ-PKAc mediated increased ERK activity and cell proliferation?

8) The reduction in pERK activation with the H33Q mutation is subtle, despite the loss of interaction with HSP70. Furthermore, there is no evidence that the association with AKAP-Lbc changes with the H33Q mutation. The MAPK recruitment interaction could occur through PKA and not HSP70; this is not addressed.

9) The proposed mechanism involves HSP70, although the function of HSP70 in this context is unclear. For example, the authors suggest (subsection “Hsp70 is recruited to DNAJ-PKAc complexes”) that the "oncogenic nature of DNAJ-PKAc….results from the recruitment of Hsp70" and in subsection “DNAJ-PKAc in FLC tumors” that the stability of the DNAJ-PKAc fusion protein may be enhanced by Hsp70? Does the DNAJ-PKAc H33Q mutant have reduced stability and impaired transforming capacity relative to DNAJ-PKAc? The authors should also comment on previous analyses of the ERK pathway that have identified HSP70 in complexes with KSR (PMC84397).

10) Figure S4 presents important data using the drug 1NM-PP1 that demonstrate that the effects of DNAJ-PKAc are not mediated by PKA activity, but these data are not described in the Results section of the manuscript. In support of Figure S4, were predicted PKA substrates from Figure 1E less phosphorylated following 1NM-PP1 treatment? The authors adapt their model from Smith et al., (2010). In that paper, the PKA in complex with AKAP-Lbc mediates phosphorylation of Ser838 on KSR1 to enhance ERK activation. This contradicts the model published in 2010 and adapted for Figure 4A. The authors should clarify this point by determining the extent to which KSR1 and Ser838 phosphorylation on KSR1 contribute to the effect of the DNAJ-PKAc fusion when it is expressed in AML12 cells.

11) Figure 1E and Figure 5D. There is no discussion of the phosphorylation sites that decrease in abundance. Does phosphorylation of some ERK substrates (or substrates of the other noted kinases) fall? If so, this would suggest an effect of the fusion protein on a shift or reprioritization of substrates rather than a mere increase in the amount of phosphorylation.

12) The inhibitors in the chemical library and the data obtained should be provided as supplemental data. A list of all the phosphorylation sites from Figure 1 and Figure 5 together with relative detection (q value and fold change) should also be included as supplemental data.

---

## [Author Response]

Essential revisions:1) Much of the Discussion section appears to be an extension of the Results section. Several figures are presented for the first time in this section. This should be corrected with all new data presented in the Results section of the manuscript.

In accordance with the reviewers’ suggestion we have expanded and modified the Discussion section. To our knowledge the only mention of “figures presented for the first time in this (Discussion) section” were the results discussed in original Figure S4. This data has been removed (see response 10 for details).

2) The study is focused on ERK. Do FLCs show evidence of increased ERK activity?

The reviewers astutely raise an important question. The seminal case report of Fibrolamellar Hepatocellular Carcinoma (FLC) noted that this liver tumor is infiltrated with fibroid bands interspersed between cancerous hepatocytes (Craig et al., 1980). This morphological feature known as “intratumoral heterogeneity” is key to answering the reviewer’s question (Pribluda et al., 2015; Liu et al., 2018).

Immunoblot analyses of FLC tumor lysates presented in a newly added figure 4A, detect a slight reduction in the overall phospho-ERK signal in patient samples. Yet immunofluorescent staining of tumor sections detects clusters of prominent and moderate phospho-ERK signal in the cancerous hepatocytes (Figure 4B and C, yellow; from “patient 3”). Such regional detection of phospho-ERK is consistent with heterogeneous intratumoral activation of the ERK cascade in FLC tumors. Likewise, the phosphoproteomic screen presented in figures 1E and F identifies numerous ERK substrates that are elevated in FLC tumors as compared to normal liver. To emphasize this point we include a list of enriched ERK substrates in figure 4D. This includes the protein kinase P90RSK, a well characterized downstream target of ERK (Dalby et al., 1998). Targeted validation of this ERK phosphorylation event in FLC tumors is provided in two ways. (1) Immunoblot detection reveals elevated levels of pSer 221-P90RSK in the same cohort of FLC tumor samples (Figure 4E). (2) Immunofluorescent detection of phospho-P90RSK in tissue sections reveals clusters of cells containing activated kinase (Figure 4F and G, magenta; example from patient 3). The heterogeneous staining profile for phospho-P90RSK shown in Figure 4F and 4G is reminiscent of the pattern of intratumoral ERK activation detected in FLC tumors (Figure 4B). Collectively these new data infer that the ERK kinase cascade is active in a subset of cells within the heterogeneous intratumoral environment of FLCs. These new data are introduced in the Results section and the Discussion section.

3) Are AML12 DNAJ-PKAc cells transformed? The observation that 15/15 FLC samples contained the fusion protein (Honeyman et al., 2014) support its role in tumor formation. It is not clear if this fusion is a driver of tumorigenesis in the model the authors created, yet the authors refer to it as oncogenic (Abstract, Introduction, subsection “Hsp70 is recruited to DNAJ-PKAc complexes”). This point is important because a major conclusion of this study is that combination therapy with Ver-155008 plus MEK inhibitor may suppress FLC. If the AML12 DNAJ-PKAc cells are not transformed, the relevance of the Ver-155008 plus MEK inhibitor combination is diminished. Although in vivo experimentation is not deemed essential for the response to this critique, it should be noted that the efficacy of the proposed drug combination using a FCL model (e.g., FCL PDX, Oikawa et al., 2015) could be tested.

The AML12^DNAJ-PKAc^ cells do not spontaneously form tumors when injected into immune compromised mice over a time course of three months (Raymond Yeung, personal communication). These properties are consistent with the very slow growth of cells from the PDX model of FLC that has been noted by other FLC investigators and discussed in our cover letter and mentioned in the Oikawa et al., article cited by the reviewers (Oikawa et al., 2015). Nevertheless, several lines of evidence indicate that the DNAJ-PKAc fusion kinase is oncogenic: (1) To further address this issue we have used colony formation assays to reinforce our data that AML12^DNAJ-PKAc^ cells have increased proliferative capacity as compared to their wildtype counterparts Figure 3D. Images and quantitation of data from multiple expeFriments is included in figure 3E. These results in subsection “Hsp70 is recruited to DNAJ-PKAc complexes”. (2) Two groups have independently used CRISPR to induce expression of the DNAJ-PKAc fusion in mice (Kastenhuber et al., 2017; Engelholm et al., 2017). In both cases, expression of DNAJ-PKAc promotes very slow growing liver tumors that share some phenotypic characteristics of FLC. One overriding limitation is that these animal models only develop tumors after two years. The tumorigenesis in old age phenotype of these mice contradicts the early onset of FLC in healthy adolescents (15-25 years). This prompted us to note in the original manuscript that “FLC research to date has been hampered by the limited availability of patient samples, a paucity of disease-relevant cell-lines, and mouse models with a long latency to develop hepatic tumors in old age”. For these reasons, our work is primarily focused on understanding the molecular mechanisms that underlie DNAJ-PKAc action in FLC. We hope that the next phase of this study will be to move our findings to GEMMs, future PDX mouse models and organoids.

4) Were the three hits from the screen the only MEK inhibitors in the screen? Were ERK or Raf inhibitors included? If so, why do the authors think they were not effective? If not, do these inhibitors provide the same level of sensitivity as the MEK inhibitors when tested in combination with Ver-155008?

Yes, cobimetinib, binimetinib and trametinib were the only MEK inhibitors included in the combination drug screen. Drugs against upstream elements of MAP kinase signaling cascades such as EGFR were also tested. The HER inhibitors poziotinib and neratinib had modest effect in combination with Ver155008. Interestingly, the lone ERK inhibitor included, GDC-0994, had little effect on proliferation in these screens. This is discussed in the Discussion section. We have included additional details on the drug screen, including the raw data for single and combination assays, as supplemental materials (see response to point 12 below).

Interestingly, our combination drug screen also included analysis of the Raf/Braf inhibitors sorafenib, dabrafenib and vemurafenib. Acting on the advice of the reviewers, we have further investigated the effects of sorafenib in combination with Ver155008. Interestingly this drug combination has no effect on cell growth at low concentrations, but paradoxically enhances cell growth at higher concentrations (300 nM – 1 µM) in wild type and AML12^DNAJ-PKAc^ cells. On the basis of these studies, we interpret the exquisite sensitivity to MEK inhibition to suggest that DNAJ-PKAc may be acting downstream of Ras-Raf activation, at the level of MEK. We provide this data Author response image 1 for the reviewers to evaluate.

5) Figure 1 shows colocalization of PKAc and RIIa. While there is some overlap in FLC there is no evidence to suggest that this colocalization is in association with the DNAJ-PKAc fusion protein versus the native PKAc since all tumors appear to retain a native PKAc. Moreover, there is no insight into the relevance of RIIa versus other regulatory subunits, including other disease associated isoforms like RIa.

There are multiple issues raised in this section:

5a) The reviewers are correct that FLC tumors retain a single allele of wild type PKA along with the DNAJ-PKAc fusion. This chimeric enzyme consists of the first 69 amino acids of Hsp40 fused in frame to residues 11 to 351 of the PKAc-α subunit. Thus, one can clearly delineate native PKA from the larger fusion kinase by immunoblot using monoclonal antibodies against PKAc (see Figure 1B, 1H, Figure 2G, 2H, Figure 5B-D and Figure 6C). However, antibodies are not available that recognize the GEEVKE junction between the DNAJ and PKAc moieties. Thus, we are currently unable to discern if PKAc and DNAJ-PKAc adopt distinct subcellular locations inside cells or tumor sections. For the record, we have tried unsuccessfully to raise antibodies against the DNAJ to PKAc junction and are currently working under the auspices of the FLC foundation to screen for monobodies that selectively recognize the fusion kinase.

5b) It is well established that DNAJ-PKAc can associate with R subunits to form functional PKA holoenzymes in FLC tumors (Honeyman et al., 2014; Riggle et al., 2016; Cao et al., 2019). In 2016 we reported that RI expression is enhanced and RIIβ levels decrease in FLC (Riggle et al., 2016). This article was cited as reference 10 in the original manuscript. As a further response to the reviewer’s request, we have incorporated new biochemical data substantiating the elevated expression of RI in FLC tumors as compared normal adjacent tissue (Figure 1—figure supplement 1A, top panels). Related experiments demonstrate that RII levels do not fluctuate (Figure 1—figure supplement 1A, bottom panel). This data is introduced in subsection “DNAJ-PKAc in fibrolamellar carcinomas” and initially discussed in the Discussion section.

5c) The reviewers consider RIa as “a disease associated isoform’. Yet, the issue of whether the type I and type II regulatory subunits of PKA fulfill distinct biological roles remains an open question. Listed below are points of clarification.

i) Physiochemically, the main differences between RI and RII is the lack of an autoregulatory phosphorylation site in RI isoforms and subtle variance in the in vitro cAMP binding affinities of both R subunits isotypes (Taylor et al., 2012).

ii) Although it was originally proposed that RII subunits were compartmentalized by AKAPs and RI was soluble (Corbin et al., 1977; Colledge, et al., 1999), many labs including our own have shown that RI subunits are sequestered by anchoring proteins. Since then, 3 out of the 42 known human anchoring protein genes SKIP, Integrin-α4 and smAKAP have been shown to selectively interact with type I PKA holoenzymes (Means et al., 2011; Lim et al., 2007; Burgers et al., 2012).

iii) There are three examples where lesions in RI subunit genes have been linked to disease. In the rare endocrine neoplasia Carney complex nonsense and insertion mutations reduce levels of RIα (Horvath et al., 2010; Stratakis, 2013). Changes in the cAMP binding sites that render RIα less sensitive to cAMP have been linked to the rare skeletal dysplasia syndrome acrodysostosis type I (Rhayem et al., 2015). Perhaps the most relevant case to our study is a single case report that inactivating mutations in RI can induce sporadic fibrolamellar carcinomas in the absence of the DNAJB1-PRKACA fusion gene (Graham et al., 2018). Although the molecular mechanism surrounding this latter case is unclear, one can postulate that loss of type I PKA, loss of anchoring to RI selective AKAPs, or overcompensation by type II PKA holoenzymes contribute to pathogenesis.

We have expanded our Discussion section to consider “the relevance of RIIa versus other regulatory subunits, including other disease associated isoforms like RI” and how changing the balance of R subunit expression might impact PKA holoenzyme signaling in FLC. We thank the reviewer for this suggestion.

6) Figure 3. The characterization of PKA activity in AML12DNAJ-PKAc lacks a statistical comparison with the parental line. Furthermore, it is unclear if the level of PKA activity in AML12DNAJ-PKAc accurately reflects that of patients (documented in Figure 1). It may be best not to minimize the importance of the PKA activity of DNAJ-PKAc based on data in the AML12DNAJ-PKAc cells without first confirming that this model reflects PKA activity in FLC.

We thank the reviewers for pointing out this missing part of the analysis and concerns about the impression that we “minimize the importance of the PKA activity of DNAJ-PKAc”. Our response is in three parts.

6a) We have decided to move our PKA activity measurements to supplemental information and have incorporated the analysis of R subunit levels suggested by the reviewers (see response 5b). This new format also allows us to clarify our views on the importance of aberrant PKA activity in etiology of FLC. Our intention is certainly not to discount the kinase activity of DNAJ-PKAc as a factor that contributes to the pathology of FLC. Rather, we have discovered that recruitment of Hsp70 via the J domain of the fusion kinase is an important new element that contributes to the dysregulation and effects this enzyme in a manner that drives oncogenesis. This point is emphasized in the Discussion section.

6b) We believe that our results, when considered with data from Riggle et al., suggests that the AML12^DNAJ-PKAc^ cells represent a valid and sustainable model system to evaluate PKA activity in FLC. For example, RI expression levels are elevated in AML12^DNAJ-PKAc^ cells in a similar manner has detected in FLC tumors (Figure 1—figure supplement 1B). Moreover, cAMP-stimulated PKA activity is potentiated in cells or tumor tissue expressing DNAJ-PKAc (Riggle et al; Figure 4).

We respect the reviewer’s point that it may be best not to minimize importance of PKA activity in FLC. In response to this suggestion we include new supplemental data (Figure 1—figure supplement 1B-D) suggesting that the pattern of PKA phosphorylation may be different in tumors and have emphasized this issue in subsection “DNAJ-PKAc in fibrolamellar carcinomas”.

6c) We provide three additional pieces of supplementary data about PKA functionality in FLC. (1) An RII overlay survey of A-kinase anchoring proteins reveals a distinct pattern of anchoring proteins in FLC tumor as compared to adjacent liver tissue (Figure 1—figure supplement 1D). This argues that the subcellular distribution of PKA and DNAJ-PKAc may be altered in FLC. This substantiates data in figures 5B-D showing increases in AKAP-Lbc expression in FLC tumors as well as the AML12^DNAJ-PKAc^ cells. (2) immunoblot data using the phospho-PKA substrates antibody detects a different pattern of PKA substrates in tumors as compared to adjacent liver extracts (Figure 1—figure supplement 1E). (3) One interpretation of this latter finding is that introduction of DNAJ-PKAc alters substrate preference or access to a different repertoire of local targets. This comports with phosphoproteomic data that identifies PKA substrates upregulated in FLC tumor samples (Figure 1—figure supplement 1F). Issues and speculation on the role of PKA activity in FLC are covered in the Discussion section.

7) The proposed mechanism involves the interaction of DNAJ-PKAc with AKAP-Lbc (Figure 4). This conclusion is speculative because no studies of AKAP-Lbc function are presented. Specifically, does loss of AKAP-Lbc decrease DNAJ-PKAc mediated increased ERK activity and cell proliferation?

First of all, we would like to point out that we can isolate the AKAP-Lbc DNAJ-PKAC subcomplex from FLC tumor and AML12^DNAJ-PKAc^ cells (Figure 5B and C). Moreover, biochemical analyses suggest this anchoring protein is upregulated in clinical samples of FLC (Figure 5B). In transient assays, we are unable detect a change in ERK activation after siRNA-mediated knockdown of AKAP-Lbc. We are currently attempting to establish both lentiviral and inducible shRNA knockdown and CRISPR knockout of AKAP-Lbc on the background of our AML12^DNAJ-PKAc^ cells. This is something we continue to develop in hopes of providing a Research Advance for this article in the future.

8) The reduction in pERK activation with the H33Q mutation is subtle, despite the loss of interaction with HSP70. Furthermore, there is no evidence that the association with AKAP-Lbc changes with the H33Q mutation. The MAPK recruitment interaction could occur through PKA and not HSP70; this is not addressed.

There are two responses to this section.

8a) We have performed additional co-immunoprecipitation experiments in AML12 cells using AKAP-Lbc as the scaffold to isolate DNAJ-PKAc-Hsp70 sub-complexes. These data show that the H33Q mutation greatly reduces the level of Hsp70 in AKAP-Lbc complexes. The simplest explanation of this result is that addition of the J-domain onto the N-terminus of PKAc induces a novel interaction with Hsp70, thereby permitting the recruitment of this chaperonin to AKAP signaling islands. This new data is presented in figure 6C and subsection “AKAP-Lbc scaffolds promote ERK activation in FLC”.

8b) We did not suggest, nor conclude that additional MAPK cascade members are recruited through Hsp70 (or through direct interaction with the PKA catalytic subunit). There is no evidence for such a mechanism. We have added narrative to the Discussion section to clarify this point.

9) The proposed mechanism involves HSP70, although the function of HSP70 in this context is unclear. For example, the authors suggest (subsection “Hsp70 is recruited to DNAJ-PKAc complexes”) that the "oncogenic nature of DNAJ-PKAc….results from the recruitment of Hsp70" and in subsection “DNAJ-PKAc in FLC tumors” that the stability of the DNAJ-PKAc fusion protein may be enhanced by Hsp70? Does the DNAJ-PKAc H33Q mutant have reduced stability and impaired transforming capacity relative to DNAJ-PKAc? The authors should also comment on previous analyses of the ERK pathway that have identified HSP70 in complexes with KSR (PMC84397).

The work of Stewart et al., (1999) is foundational in the understanding of KSR function. We have now cited this work in the revised manuscript. As the reviewer is undoubtedly aware, this article predominantly focuses on evidence that MEK is a component of this complex, as well as identifying HSP90/cdc37 and Hsp70 interactions with the KSR scaffold. This work was conducted a decade before our discovery of the AKAP-Lbc-KSR signaling unit, and at a time when the sensitivity of protein identification within signaling scaffolds was much less sophisticated than the quantitative mass spectrometry approaches today. Nonetheless, these findings of Stewart et al. remain in agreement with the conclusion in Figure 5A-E that the chaperonins including the Hsp70-DNAJ-PKAc subcomplex are only recruited to AKAP-Lbc-KSR signaling units in FLC. As per reviewer suggestion we have added a note in the Discussion section acknowledging that Hsp90 and certain Hsp70 isoforms may interact with KSR via other mechanisms.

10) Figure S4 presents important data using the drug 1NM-PP1 that demonstrate that the effects of DNAJ-PKAc are not mediated by PKA activity, but these data are not described in the Results section of the manuscript. In support of Figure S4, were predicted PKA substrates from Figure 1E less phosphorylated following 1NM-PP1 treatment? The authors adapt their model from Smith et al., (2010). In that paper, the PKA in complex with AKAP-Lbc mediates phosphorylation of Ser838 on KSR1 to enhance ERK activation. This contradicts the model published in 2010 and adapted for Figure 4A. The authors should clarify this point by determining the extent to which KSR1 and Ser838 phosphorylation on KSR1 contribute to the effect of the DNAJ-PKAc fusion when it is expressed in AML12 cells.

We thank the reviewer for these comments. There are two responses to this section.

10a) We have reevaluated the data with the analog sensitive DNAJ-PKAc M176A and now propose an alternative explanation as to why application of 1NM-PP1 has no apparent effect on ERK activation. Since PKA holoenzymes are dimers of dimers, AKAP-Lbc has the capacity to simultaneously sequester one copy of wildtype PKAc with an analog sensitive DNAJ-PKAc ortholog in the context of active and intact holoenzymes (Smith et al., 2017). Under this scenario, only one element of the anchored PKA holoenzyme would be sensitive to 1NM-PP1 and regulation of KSR mediated activation of the ERK cascade may still proceed. Due to possibility of this alternate interpretation we have removed the analog data and thank the reviewer for their astute observations that led us to this outcome.

10b) Our published work shows that anchored PKA promotes phosphorylation of Ser838 on KSR to support cAMP responsive activation of the RAF-MEK-ERK cascade(Smith, et al., 2010). Recent work confirms that S838 on KSR is an important regulatory site for PKA that influences association with BRAF (Takahashi, et al., 2017). Experimental support for our model of DNAJ-PKAc regulation of KSR phosphorylation and ERK activity is found in the phospohproteomic data presented in Figure 1 and Figure 1—figure supplement 1F. This data revealed elevated phosphorylation of the PKA site serine 838 on KSR in FLC tumors. Thus, we propose that DNAJ-PKAc is promoting KSR phosphorylation to support MAPK pathway activation. Unfortunately, we do not have antibodies that can specifically recognize phospho-Ser838 in KSR. Mention of this new information along with a more general statement recognizing the potential role of PKA activity as a pathological factor in the development of FLC in now included in the Discussion section.

11) Figure 1E and Figure 5D. There is no discussion of the phosphorylation sites that decrease in abundance. Does phosphorylation of some ERK substrates (or substrates of the other noted kinases) fall? If so, this would suggest an effect of the fusion protein on a shift or reprioritization of substrates rather than a mere increase in the amount of phosphorylation.

The reviewers are correct to point out that phosphoproteomic analysis always detects changes in phosphorylation sites in both directions. We initially omitted discussion of sites that decrease for the sake of simplicity. We now include an analysis of bi-directional changes in phosphorylation sites for the normal vs tumor data (Figure 1—figure supplement 2) in subsection “DNAJ-PKAc in fibrolamellar carcinomas” and subsection “AKAP-Lbc scaffolds promote ERK activation in FLC”.

In addition, we are attaching an Excel spreadsheet with all the phosphoproteomic data from the tumors and AML12^DNAJ-PKAc^ and AML12^WT^ cells. This data shows the collective phosphorylation events that change in both directions for these analyses (see also response to point 12).

12) The inhibitors in the chemical library and the data obtained should be provided as supplemental data. A list of all the phosphorylation sites from Figure 1 and Figure 5 together with relative detection (q value and fold change) should also be included as supplemental data.

The chemical library screen data and the phosphoproteomic data is now included as supplemental Excel spreadsheets.